# Mask Based Unsupervised Content Transfer

Ron Mokady[1], Sagie Benaim[1], Lior Wolf[1,2], and Amit Bermano[1]

[1]The School of Computer Science, Tel Aviv University
[2]Facebook AI Research

## Abstract

We consider the problem of translating, in an unsupervised manner, between two domains where one contains some additional information compared to the other. The proposed method disentangles the common and separate parts of these domains and, through the generation of a mask, focuses the attention of the underlying network to the desired augmentation alone, without wastefully reconstructing the entire target. This enables state-of-the-art quality and variety of content translation, as demonstrated through extensive quantitative and qualitative evaluation. Our method is also capable of adding the separate content of different guide images and domains as well as remove existing separate content. Furthermore, our method enables weakly-supervised semantic segmentation of the separate part of each domain, where only class labels are provided. Our code is available at `https://github.com/rmokady/mbu-content-tansfer`.

## 1 Introduction

The task of content transfer, as depicted in Fig. 1, involves identifying the component of interest (for example, glasses) in a given input (for example, an image of a face), adapting it, and adding it to a second given input (for example, another image of a face, without glasses), hopefully in the semantically correct manner. Such an operation can be used to prototype or demonstrate changes in appearance Gatys et al. (2016), augment music Grinstein et al. (2018), compose text Prabhumoye et al. (2019), generate data for training purposes Mueller et al. (2018), etc.

Recent advancements (Huang et al. (2018); Lee et al. (2019)) translate one domain to another with varying styles, but not content. Others (Lample et al. (2017); He et al. (2019); Liu et al. (2019)) produce images in a target domain with a given attribute (e.g glasses), but such attribute is unique and not varying or controlled (e.g., not allowing the specification of a specific pair of glasses).

A recent advancement in the realm of attribute transfer has been presented by Press et al. (2019). In this work, the input is two domains of images, such that the images in one domain, $B$, contain a specific class (e.g. faces with facial hair), while in the other domain, $A$, the images do not (e.g., faces without facial hair). Training on this input, the method learns to transfer only the specific class information from an unseen image in the domain $B$ to an unseen one in the domain $A$, while preserving all other details. The proposed architecture yields a simple network which is able to perform the required disentanglement through an emergence effect. In this setting, the content to be added by the system is not explicitly marked in the target domain, nor does it have an equivalent counterpart in the source domain for training (e.g. an image with and without glasses of the same person). A form of weak supervision can be provided by a simple annotation of whether the relevant content exists or not in every example. However, this method, and other typical ones addressing similar tasks, generate the details for the entire image, by using auto-encoder or GAN-based architectures, resulting in a degradation of details and quality.

In this paper, we build upon the emerging disentanglement idea, but also adopt the growing understanding that one should minimize redundant use of computational resources and model parameters (Chen et al. (2016); Mejjati et al. (2018); Chen et al. (2018)), to the aforementioned task. In other words, using a mask, we focus the attention of the baseline network to the desired augmentation alone, without asking it to wastefully reconstruct the entire target. As can be seen in Fig 2, the method consists of two main steps. The first is the disentanglement step, which encodes the domain specific

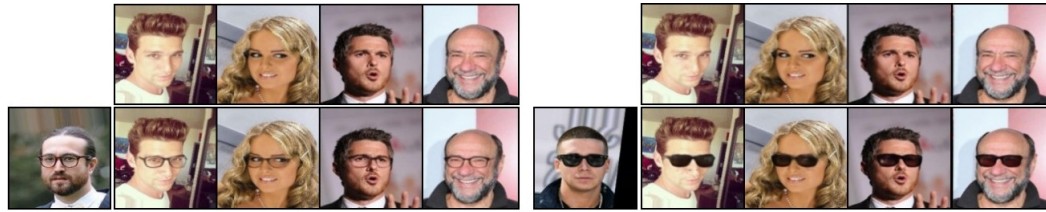

Figure 1: Content transfer example. Given an image of a face with glasses (left), and another image of a face without ones (top), the proposed method successfully identifies and translates the specified glasses from the former domain to the latter one.

and the domain invariant contents separately, and is inspired by the work of Press et al. (2019). The second step is the key insight of our proposal. It locates the part of the target that should be changed and generates relevant augmentation content to go with it. This allows keeping the unrelated details intact, facilitating a great improvement in generation quality; The augmentation focuses on the relevant part, leaving all other details to be taken from the target image directly, without going through the bottleneck of an auto-encoder-like module.

By applying this simple yet effective principle and a novel regularization scheme, our method preserves target details which are irrelevant to the augmentation and is able to improve upon the state-of-the-art in terms of quality and variety of the content transferred. Furthermore, we demonstrate how the method performs well, even when presented with images outside the domain trained on, and that the aforementioned mask, generated by the system to mark the regions of augmentation, is accurate enough to provide a semantic segmentation of the content transferred.

Lastly, our method can also be used in the opposite direction — to remove the domain specific attribute, and thus translating from $B$ to $A$. Despite not being our main focus, we outperform the various literature methods that address this task (Lample et al. (2017); Liu et al. (2019); He et al. (2019); Press et al. (2019)). Our advantage is also evident in that once an object is removed, we can then add a different attribute to the resulting image, thus allowing the translation from any two domains, each having a domain specific attribute (e.g removing glasses and adding the smile of a guide image). A related unique ability is that of removing and adding the same attribute, e.g. replacing one's facial hair with a different facial hair.

## 2  PREVIOUS WORK

In the unsupervised image to image translation, the learner is given two unpaired domains of visual samples, $A$ and $B$, and is asked, given image $a \in A$ to generate an analogue image in the domain $B$. This problem is inherently ill-posed, as multiple analogous solutions may exist. In several of different approaches (Zhu et al. (2017a); Kim et al. (2017); Yi et al. (2017)) a circularity constraint is used to reduce this ambiguity. COGAN Liu & Tuzel (2016) and UNIT Liu et al. (2017) enforce a shared latent representation between the two domains. Unlike our method, these methods produce a single solution per input image $a$.

Moving from one to one mappings, multiple approaches provide many to many mappings. Supervised multimodal approaches, where paired samples are provided, include BicycleGAN Zhu et al. (2017b), which injects random noise $z$ in a generator and enforces an encoder to recover $z$ from the target translation, and MAD-GAN Ghosh et al. (2018). The latter trains multiple generators to produce aligned mappings, which are distant from each other. These methods require paired samples from both domains — a costly supervision, which we do not require.

**Guided Multimodal Approaches** MUNIT Huang et al. (2018), DRIT Lee et al. (2018), and DRIT++ Lee et al. (2019) are trained on unmatched images. MUNIT trains two encoders; one captures the content of an image, and another its style, inducing disentanglement. During inference, multiple solutions are produced using the style of a guide image in the target domain. In DRIT (and DRIT++), a cycle constraint is employed, in a setting where the generator of each domain consists of two disentangled encoders, one of which encodes the content and the second the style of the image. For all these methods, different non symmetrical architectures of encoders are used to capture the

style and content. In MUNIT, for example, residual connections are used for the content encoder, and global pooling and adaptive instance normalization are used for the style encoder. Hence, the style code is significantly smaller in dimensions than the content one. In our method we employ two encoders as well, but the architecture of these encoders is symmetric, allowing both encoders to capture content in both.

The most relevant work to ours is that of Press et al. (2019), which also uses the setting in which the samples in domain $B$ contain all the information in domain $A$ and some additional information. Two encoders are used — the first captures the information that is common between the two domains and the second encodes the unique information of domain $B$. The decoder maps the concatenation of the two encodings into an image in the domain $B$, or, in the case that the second encoding is set to zero, to an image in $A$. Content is transferred between images by mixing the encoding of the former type of one image with the encoding of the latter type of a different image.

In many cases, however, only a local area in the image needs to change during translation. Consider the case where $A$ is images of faces and $B$ is faces with facial hair. For $a \in A$, only the location in which the facial hair is placed in $a$ needs to change. In the method of Press et al. (2019), the entire image, including other facial features, is generated from scratch, and as a result, many low level details are lost and the quality of generation is reduced. This is not the case for our method, where outside the generated mask, which denotes the location of the facial hair, the content of the generated image is taken from the input image $a$. This is achieved by employing two decoders, one for domain $A$ and one, with two outputs (raw image and mask) in domain $B$, and by a new set of loss terms.

**Mask Based Approaches**    The use of masks is prevalent in a variety of visual tasks. For example, for virtual try on, Han et al. (2018) uses a supervised human parser network to transfer the desired clothing to a target person. Unlike our method, this method cannot perform general content transfer as it crucially relies on a supervised pose estimator. In the context of style transfer, Ma et al. (2018) uses masks to transfer the style of semantically similar regions from the target to the source image. In the context of image to image translation, the one to one case was addressed by Chen et al. (2018); Mejjati et al. (2018), where in addition to mapping to the target domain, a mask is learned, to cover only the relevant area in the translation. For example, in the case of mapping from horses to zebras, the mask learns to cover the area of the zebra, which allows the background to be taken from the source image, thus allowing for a much better quality of generation. In our method, we extend this masking (or attention) approach to the one to many (guided) case. Note that while Chen et al. (2018); Mejjati et al. (2018) learn a mask and then employ it directly to the image, this does not allowthe mask to adapt both target and guide image.

**Weakly supervised semantic segmentation methods** can be stratified based on the type of supervision used. In the first set of methods (Song et al. (2019); Hu et al. (2018); Zhao et al. (2018)) a bounding box is used. Other approaches (Cheng et al. (2018); Caelles et al. (2017)) use the entire supervision of the fully supervised approach, but are required to find a segmentation in a single shot. Our approach belongs to a set of methods Zhang et al. (2018); Zhou et al. (2018); Wei et al. (2018); Ahn & Kwak (2018) that use only the class label information to find a segmentation. Zhou et al. (2018) use the visual cues arising from peaks in class response maps (local maxima) to generate highly informative regions. Zhou et al. (2016) use Class Activation Maps (CAMs) extracted from a classifier to obtain discriminative localization. Wei et al. (2018) use varying dilation rates to transfer surrounding discriminative information to non-discriminative object locations. Ahn & Kwak (2018) propagate local discriminative parts to nearby regions that belong to the same semantic entity. Other methods (Tang et al. (2018b); Kervadec et al. (2019); Tang et al. (2018a)) use regularized losses with different levels of supervision. Unlike these methods, our main focus is on generating the added part in a way that it is adapted to the placement context for the domain specific information and yet, our segmentation results are competitive with such methods on this task.

## 3 METHOD

We transfer content that exists in a sample $b$ in domain $B$ onto a sample $a$ from a similar domain $A$, in which this *domain specific* content is not found. In addition, we also consider the task of weakly supervised semantic segmentation of the domain specific content. That is, given unpaired samples from domains $A$ and $B$, we wish to label (or generate a segmentation mask for) the domain specific

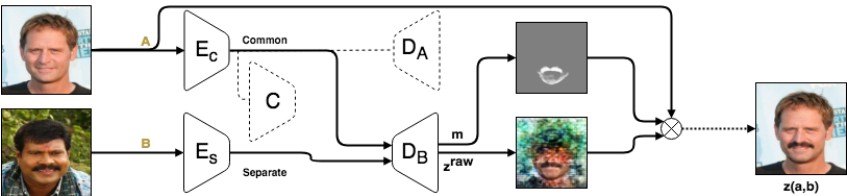

Figure 2: An illustration of the inference procedure. The discriminator $C$ and decoder $D_A$ are not used during inference but are included for illustrative purposes.

part in an image $b \in B$. We also consider attribute removal. That is, given $b \in B$, we wish to remove the domain specific part of $b$.

Our method consists of five different networks: the common encoder, $E_c$, aims to capture the *common* (or domain invariant) information between domains $A$ and $B$. The separate encoder, $E_s$, aims to capture the *separate* (or domain specific) information in domain $B$. The domain confusion network, $C$, is used to make the encodings generated by $E_c$ for images from both domains indistinguishable. The decoder, $D_A$, generates samples in domain $A$, given a representation that is obtained by the common encoder $E_c$. If that sample comes from domain $B$, the domain-specific content is removed.

The generation of the image that combines the content of $a$ and the domain specific content of $b$ is done by the decoder $D_B$, which returns two image-sized outputs: $z^{\text{raw}}$ and $m$.

$$m(a,b), z^{\text{raw}}(a,b) = D_B(E_c(a), E_s(b)) \tag{1}$$

where $m(a,b)$ is a soft mask with values between 0 and 1 and $z^{\text{raw}}$ is an image. It is important to note that the mask and the generated image both depend on the content in $b$ as well as on the image $a$, which determines the placement and other appearance modifications.

The final output $z$ is a combination of these outputs and image $a$,

$$z(a,b) = m(a,b) \otimes z^{\text{raw}}(a,b) + (1 - m(a,b)) \otimes a, \tag{2}$$

where $\otimes$ stands for an element wise multiplication. Fig. 2 illustrates the inference step as well as the five networks.

**Domain Confusion Loss.** We seek to ensure that the common encoding, generated by $E_c$, contains only information that is common to both domains. This is done by combining reconstruction losses with a domain confusion loss. The latter employs a discriminator network, $C$, that encourages the encodings of the two domains to statistically match Ganin et al. (2016).

$$\mathcal{L}_{DC} := \frac{1}{|S_A|} \sum_{a \in S_A} l(C(E_c(a)), 1) + \frac{1}{|S_B|} \sum_{b \in S_B} l(C(E_c(b)), 1) \tag{3}$$

where $S_A$ and $S_B$ are the training sets sampled from the two domains and $l(p,q) = -(q \log(p) + (1-q) \log(1-p))$ is the binary cross entropy loss for $p \in [0,1]$ and $q \in \{0,1\}$.

Our formulation of the domain confusion loss is similar to that of Tzeng et al. (2017) except where for $\mathcal{L}_{DC}$, $E_c$ attempts to fool the discriminator $C$, so that the encodings of both domain A and domain B would be classified as 1. Namely, $C$ tries to distinguish between encodings of domain A and B, while $E_c$ attempts to produce an encoding which is indistinguishable for $C$.

While $E_c$ attempts to make the two distributions indistinguishable, $C$ is trained in an adversarial manner to minimize the following objective:

$$\mathcal{L}_C := \frac{1}{|S_A|} \sum_{a \in S_A} l(C(E_c(a)), 0) + \frac{1}{|S_B|} \sum_{b \in S_B} l(C(E_c(b)), 1) \tag{4}$$

**Reconstruction Loss** The domain confusion loss ensures that the common encoder, $E_c$, does not encode any separate information from domain $B$. For samples $a \in A$, we also need to verify that the

information in $E_c(a)$ is sufficient to reconstruct it, ensuring that all the information of domain $A$ is encoded by $E_c$. We use

$$\mathcal{L}_{Recon1}^{A} := \frac{1}{|S_A|} \sum_{a \in S_a} \|D_A(E_c(a)) - a\|_1 \tag{5}$$

where $\|\|_1$ is the L1 loss directly applied to the RGB image values.

Similarly, we wish to verify that the information encoded by $E_s$ is sufficient for reconstructing the separate details, so that $E_s(B)$ contains the domain specific information of domain $B$. Given an image $b \in B$, we do this by removing the separate information from it, using $D_A(E_c(b))$, and adding it back:

$$\mathcal{L}_{Recon1}^{B} := \frac{1}{|S_B|} \sum_{b \in S_B} \|z'(D_A(E_c(b)), b, b) - b\|_1, \tag{6}$$

where $z'$ is defined as:

$$z'(c, a, b) = m(a, b) \otimes z^{\text{raw}}(a, b) + (1 - m(a, b)) \otimes c \tag{7}$$

For Eq. 6, we use $z'$ instead of $z$. This is so $E_c$ is not applied on $D_A(E_c(b))$, but directly on $b$. In both cases, one recovers the common information of $b$, but when using $z$, additional error is introduced thought the use of $D_A \circ E_c$.

Finally, we reinforce the roles of the two domains by encouraging the mask to be minimal. In our experiments, we saw that explicitly penalizing the mask size, or using other traditional regularization terms, yielded inferior results, as shown in Sec. 4.1. Instead, we achieve this goal in a softer way, by running samples from each domains through both inputs of our transfer pipeline and favouring successful reconstruction:

$$\mathcal{L}_{Recon2}^{A} := \frac{1}{|S_A|} \sum_{a \in S_A} \|z(a, a) - a\|_1 \qquad \mathcal{L}_{Recon2}^{B} := \frac{1}{|S_B|} \sum_{b \in S_B} \|z(b, b) - b\|_1 \tag{8}$$

The first term of the loss introduced in Eq. 8 ($\mathcal{L}_{Recon2}^{A}$) encourages a minimal distance between $z(a, a)$ and $a$, where $z(a, a) = z^{raw}(a, a) \otimes m(a, a) + a \otimes (1 - m(a, a))$. Ideally, $z^{raw}$ would be equal to $a$, but since we use an encoder and a decoder which cannot auto-encode perfectly, we get that there is some distance between $z^{raw}$ and $a$. Hence, in order to minimize the distance between $z(a, a)$ and $a$, the network minimizes the size of the mask. Similar argument holds for $\mathcal{L}_{Recon2}^{B}$.

**Cycle Consistency Losses**    Cycle consistency in the latent spaces is used as an additional constraint to encourage disentanglement. Specifically, we have:

$$\mathcal{L}_{Cycle} := \frac{1}{|S_A||S_B|} \sum_{a \in S_A, b \in S_B} \|E_c(z(a, b)) - E_c(a)\|_2 + \|E_s(z(a, b)) - E_s(b)\|_2 \tag{9}$$

where $\|\|_2$ is the MSE loss.

The overall loss term we minimize is:

$$\mathcal{L} = \mathcal{L}_{DC} + \lambda_1 \mathcal{L}_{Recon1}^{A} + \lambda_2 \mathcal{L}_{Recon1}^{B} + \lambda_3 \mathcal{L}_{Cycle} + \lambda_4 \mathcal{L}_{Recon2}^{A} + \lambda_5 \mathcal{L}_{Recon2}^{B}$$

where $\lambda_1, \ldots, \lambda_5$ are positive constants. We train a discriminator $C$ separately to minimize $L_C$.

**Inference**    The network's architecture is provided in appendix A. Once trained, the networks can be used for unsupervised content transfer and weakly supervised segmentation of the domain specific information. In the first case, we generate examples $z(a, b)$ for $a \in A, b \in B$. In the second, we consider the mask generated by feeding an image $b$ from domain $B$ to **both** inputs $m(b, b)$, then apply a threshold to get a binary mask. As shown in appendix Fig. 37 the method is largely insensitive to the exact value of the threshold.

The network can also be used for attribute removal by generating $z_{unmasked} := D_A(E_c(b))$. $z_{unmasked}$ is $b$ with its separate part removed. In order to avoid missing reconstructed facial features the generated output is calculated as:

$$m(b, b), z := D_B(E_c(b), E_s(b))$$
$$z_{removed} := (1 - M(b, b)) \otimes b + M(b, b) \otimes z_{unmasked}.$$

where M is the binarized mask of the soft mask m.

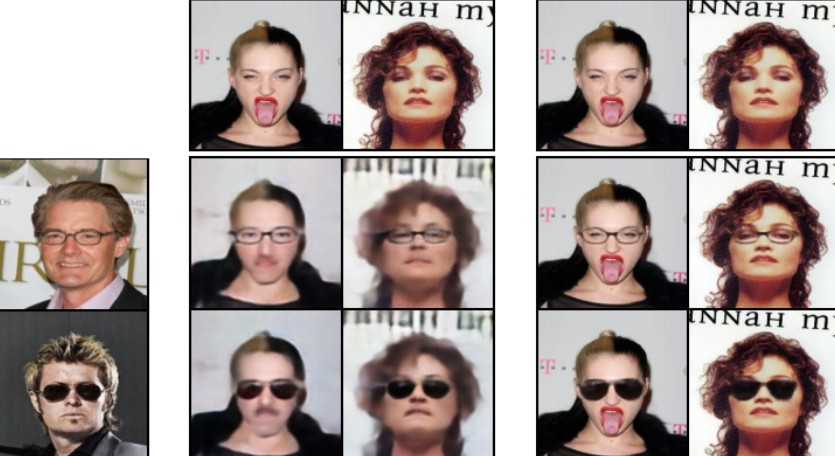

Figure 3: Glasses from guide images in domain $B$ (left) augment the glasses-less source images from domain $A$ (top). The content transfer of Press et al. (2019) (middle) is compared to our results (right).

## 4 EXPERIMENTS

We evaluate our method for guided content transfer, out of domain manipulation, attribute removal, sequential content transfer, sequential attribute removal and content addition, and weakly supervised segmentation of the domain specific content.

**Guided Content Transfer**     We employ three attributes that are expressed locally in the images of the celebA dataset Yang et al. (2015): smile, facial hair, and glasses. In each case, we consider $B$ to be the domain of images with the attribute, and $A$ to be the domain without it.

We first consider the ability to add the separate part of an image $b \in B$ to the common part of $a \in A$. This is shown for the domain of glasses, in Fig. 3, compared to the baseline method of Press et al. (2019). As can be seen, only the local structure of the glasses is changed, whereas in the baseline many low level details are lost (for example, the background writing) and unnecessary changes are made (for example, an open mouth is replaced with a closed one, or facial hair is added, changing the identity of the source image). Furthermore, Fig. 1 demonstrates the ability of our method to accommodate for different orientations of the source image $a$, and to properly adapt the glasses from $b$ to the correct orientation. Please refer to the appendix B for more examples.

To assess the quality of the domain translation, we conduct a handful of quantitative evaluations. In Tab. 1, we consider the Frechet Inception Distance (FID) Heusel et al. (2017) and Kernel Inception Distance (KID) Bińkowski et al. (2018) scores of images with the common part of $a$ and separate part of $b$ over a test set of images from domains $A$ and $B$. The FID score is a commonly used metric to evaluate the quality and diversity of produced images; KID is a recently proposed alternative for FID. We note that these values should only be used comparatively, as the size of the test set used affects the score magnitude. As can be seen, our method scores significantly better.

We also consider the ability of our method to transfer the separate part of $b$ to the target image. To do so, we use a pretrained classifier to distinguish between domains $A$ and $B$ (on the respective training sets) and consider the score of the translated images. These results are reported in Tab. 2, and show a clear advantage to our method. As expected, the MUNIT and DRIT methods Huang et al. (2018); Lee et al. (2018) are not competitive in this metric, since they transfer style and not content. Additionally, in contrast to our method, Fader networks (Lample et al. (2017)) transfer to $B$ without the use of a specific guide image $b$ from this domain.

To evaluate if the source identity is preserved, we compute the cosine similarity of the pretrained VGG-face network Cao et al. (2017). High values indicate preserved identity. Tab. 3 indicates that our results exhibit a much better similarity to source images than baseline methods.

Table 1: FID and KID scores (lower is better) for generated images using the common part of $a \in A$ and the separate part of $b \in B$. As real images, we consider the images in $A$. For KID we used $\gamma = 0.01$, kernel $k(x, y) = (\gamma x^T y + 1)^3$.

| | Facial hair | | Glasses | |
|---|---|---|---|---|
| Method | FID | KID | FID | KID |
| Real images | 85.4±2.9 | 2.5±0.2 | 115.5±3.8 | 0.1±0.3 |
| Ours | 90.7±1.8 | 3.5±0.1 | 134.9±4.8 | 5.2±0.8 |
| Press et al. | 139.4± 1.9 | 16.8±0.5 | 178.5±3.2 | 14.6±1.2 |

Table 2: The accuracy of generated images according to a pretrained classifier distinguishing between $A$ and $B$.

| | Smile | Glasses | Beard |
|---|---|---|---|
| Fader | 93.9 % | 93.6% | 81.8% |
| Press et al. | 98.9% | 94.8% | 88.1% |
| MUNIT | 8.5% | 8.3% | 7.2% |
| DRIT | 9.2% | 7.4% | 6.5% |
| Ours | 99.2% | 96.2% | 88.0% |

Table 3: An evaluation of the cosine similarity (higher is better) before and after translation between the VGG-face descriptors. Shown are average results over 100 random images created by sampling $a$ and $b$ from the test sets.

| | A to B mapping | | | Transfer A' to B' | |
|---|---|---|---|---|---|
| | Facial hair male to male | Glasses all genders | Smile all genders | Facial hair male to female | Glasses train women, test men |
| Ours | 0.89 | 0.84 | 0.94 | 0.90 | 0.82 |
| Press et al. (2019) | 0.73 | 0.68 | 0.73 | 0.64 | 0.59 |

Table 4: User study (questions (1), (2) and (3)) showing preference to our method vs. Press et al. (2019), see text.

| | A to B mapping | | | | | | | A' to B' shift | |
|---|---|---|---|---|---|---|---|---|---|
| | Facial hair male to male | Glasses all genders | Smile all genders | Hand-bags | Two Attrs | Remove Smile Add Glasses | Facial hair swap | Facial hair female $A', B'$ | Glasses train women test men |
| (1) | 96% | 95% | 55% | 87% | 91% | 83% | 93% | 91% | 93% |
| (2) | 84% | 82% | 43% | 72% | 93% | 90% | 83% | 70% | 86% |
| (3) | 97% | 95 % | 95% | 90% | 95% | 91% | 91% | 91% | 97% |

To evaluate the interpretability of the latent space, we interpolate between the latent code of the separate parts of images $b_1 \in B$ and $b_2 \in B$ with the common latent code of an image $a \in A$. This is shown in Fig. 4 and appendix C. Note the mask changes throughout the interpolation.

**User study** To further strengthen the evaluation, we conduct a user study. We randomly sample 20 images from $a \in A$ and $b \in B$ and consider the translated image of our method vs. that of Press et al. (2019). We conduct three experiments where the user is asked to select: (1) the translated image that matches the distribution of $B$ more closely, (2) Given the guide image $b$, in which translated image, the separated part of $b$ is better transferred, and (3) Given the source image $a$, which translated image better preserves the facial features of $a$. Average scores are reported in Tab. 4. For the tasks of facial hair and glasses, we score consistently higher than the baseline method. For smile, our ability to produce realistic smiling faces is slightly higher, the ability to transfer the smile from the source image is slightly worse, while our ability to preserve the identity of the source image is significantly higher. This probably stems from the smile taking place not only in the specific mouth region.

**Out of domain manipulations** We also consider the ability of the learned model handle a domain shift, i.e. to perform a translation from a domain which was not seen during training. For example, we train on female faces without glasses as domain $A$, and female faces with glasses for domain $B$. At test time, $A$ is replaced with a domain $A'$ of male faces, and we are asked to transfer the glasses onto the male's face, generating a domain $B'$ from which we see no train or test samples. Quantitative evaluation is provided in Tab. 3, showing a negligible difference in quality for our method, and a significant one for the baseline method. Visual results can be found in appendix C, where we also consider out of domain LFW dataset Huang et al. (2007) as well as extremely out of domain images, which our method successfully handles.

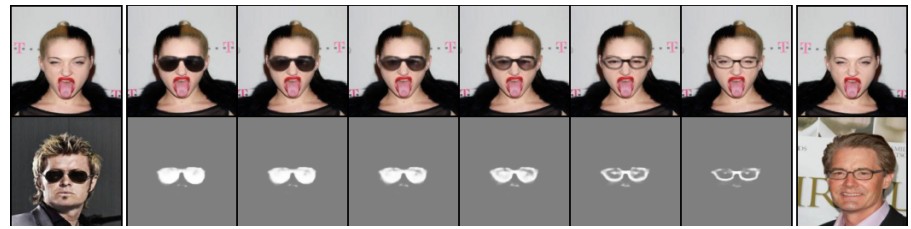

Figure 4: Interpolation between $E_s(b_1)$ (bottom left) and $E_s(b_2)$ (bottom right) for $b_1, b_2 \in B$, while fixing the source image $a \in A$ (top ends). The generated images (top) and masks (bottom) are shown.

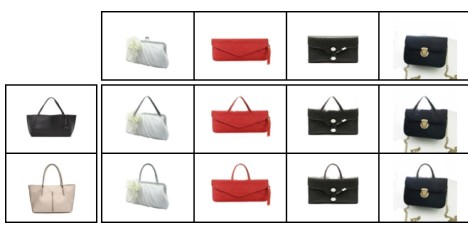

Figure 5: Adding a handle to a handbag.

Table 5: Mean and SD IoU for the two hair segmentation benchmarks.

| Method | Women's hair | Men's hair |
|---|---|---|
| Ours | $0.77 \pm 0.15$ | $0.77 \pm 0.13$ |
| Press et al. | $0.67 \pm 0.13$ | $0.58 \pm 0.11$ |
| Ahn & Kwak. | $0.54 \pm 0.10$ | $0.52 \pm 0.10$ |
| CAM | $0.43 \pm 0.09$ | $0.56 \pm 0.07$ |

**Handbags**  We also consider the domain of handbags Zhu et al. (2016), where we split this domain into images with a handle ($B$) and those without ($A$). The transfer results are illustrated in Fig. 5. The generated mask and raw outputs are clearly adapted to the bag on which the handle content is placed. The user study in Tab. 4 evaluates these results. Please refer to appendix D for visual comparison.

**Attribute removal**  While our method is more general than attribute transfer methods, it can be used to remove a given attribute as shown in Fig 6; see appendix E for a full qualitative comparison to the literature methods. A quantitative evaluation is given in Tab 6. For generation quality we use KID and FID; for successful attribute removal, a pretrained classifier is used to measure the percentage of test images without the attribute, and for similarity with the source image, a perceptual loss is used (using the features of a VGG-face Cao et al. (2017) network). Our method is significantly superior in terms of generation quality over all baseline methods for all tasks and presents a good tradeoff between fidelity and transformation success. For facial hair removal, Press et al. (2019) and Lample et al. (2017) are superior in terms of classifier accuracy, yet their generation quality is far worse (blurry images) and the similarity to the source image is significantly impaired. As the comparison images in appendix E show, Lample et al. (2017) achieves higher accuracy by producing female images while Press et al. (2019) makes the persons younger looking. He et al. (2019) has slightly superior similarity score, but is worse on removing the facial hair and has worse generation quality. For smile, Liu et al. (2019) is slightly superior in removing the smile in terms of accuracy, yet worse on generation quality and similarity to the source image. For glasses, all classifier scores are close to 100% meaning an almost perfect glasses removal, yet our similarity score and generation quality is higher.

**Sequential content transfer**  Our method enables a sequential addition of guided content from different guide images and from different domains by applying our method sequentially. Fig 28 considers this case for glasses and facial hair addition. Our method significantly outperforms Press et al. (2019), as it does not wastefully reconstruct the facial features twice as shown in Tab 4 ("two attributes") for adding facial hair and glasses.

**Attribute removal and content addition**  Given the ability of our method to remove a specific attribute, one can perform guided content transfer between any given domain, A and B, each with it separate domain specific information. First, we remove the domain specific attribute of domain A and then perform guided content addition for domain B. For example, in Fig. 4 smile is removed and glasses are then added, see appendix F for more results, as well as facial hair swap in appendix Fig 32. As for sequential content transfer, we do not wastefully reconstruct the facial features and so significantly outperform Press et al. (2019) as can be seen in Tab 4 for the task of smile removal and glasses addition as well as facial hair swap (removing and adding facial hair).

Figure 6: Attr removal.

Table 6: Attribute removal for the task of Smile, Facial hair and Glasses.

| Task | Method | KID | FID | Class. | Sim. |
|------|--------|-----|-----|--------|------|
| Smile | Ours | $2.6 \pm 0.4$ | $120.0 \pm 2.6$ | 96.9% | 0.96 |
| | Press et al. | $15.0 \pm 0.6$ | $167.7 \pm 0.3$ | 96.9% | 0.81 |
| | He et al. | $4.1 \pm 0.4$ | $127.7 \pm 4.5$ | 96.9% | 0.95 |
| | Liu et al. | $4.3 \pm 0.3$ | $129.0 \pm 3$ | 98.4% | 0.92 |
| | Fader | $11.3 \pm 0.7$ | $155.6 \pm 4.7$ | 93.7 % | 0.89 |
| Mustache | Ours | $1.9 \pm 0.5$ | $119.0 \pm 0.8$ | 95.3 % | 0.95 |
| | Press et al. | $16.6 \pm 0.8$ | $175.9 \pm 1.4$ | 100.0% | 0.80 |
| | He et al. | $4.6 \pm 0.5$ | $130.0 \pm 3.0$ | 87.5% | 0.96 |
| | Liu et al. | $14.0 \pm 0.6$ | $160.0 \pm 3.3$ | 87.5% | 0.85 |
| | Fader | $14.1 \pm 0.6$ | $162.6 \pm 1.5$ | 98.4 % | 0.76 |
| Glasses | Ours | $5.2 \pm 0.5$ | $136.5 \pm 2.6$ | 99.2% | 0.87 |
| | Press et al. | $15.3 \pm 0.5$ | $172.0 \pm 4.7$ | 100.0% | 0.73 |
| | He et al. | $8.3 \pm 0.9$ | $141.4 \pm 6.8$ | 100.0% | 0.84 |
| | Liu et al. | $6.8 \pm 0.3$ | $141.8 \pm 4.8$ | 98.4% | 0.86 |
| | Fader | $12.5 \pm 0.3$ | $137.7 \pm 4.2$ | 100.0% | 0.76 |

Figure 7: Removal of smile and addition of glasses according to the guided image on the left. In the middle, the translation of Press et al. (2019) and on the right, our result.

**Weakly supervised segmentation** We consider the task of segmenting women's and men's hair. For men, $A$ consists of bald men, while $B$ contains men with dark hair. For women, $A$ consists of women with blond hair, while $B$ contains women with black hair. We evaluate our method using the labels given in Borza et al. Borza et al. (2018).

We generate the segmentation using the method described in Sec. 3. We compare our method to Press et al. (2019), where we take the translated image and subtract, in pixel space, the source image from it. We also compare to the results obtained by the recent weakly supervised segmentation method of Ahn and Kwak Ahn & Kwak (2018), which performs segmentation using the same level of supervision we employ, using published code. In addition, we compare to CAM Zhou et al. (2016), where we train an Inception-V3 network to classify between the domain and extract localization from the classifier, which we then binarize to get a segmentation mask.

As can be seen in Fig. 8, our results provide smooth labeling of the hair, while Press et al. (2019) yield a broken one with unnecessary details. The result of Ahn & Kwak (2018) also lacks in comparison. CAM is unable to generate the required shape, as the classifier always focuses on the same place. Similar results are shown for man's hair in the appendix G. Our results are also superior quantitatively, as shown in Tab. 5 for the Intersection over Union (IoU) measure. We also perform semantic segmentation for both glasses and facial hair, refer to appendix G. The success of our method stems from requiring the network to minimally add the separate content in the correct location to reconstruct $b \in B$, thus localizing the separate content.

## 4.1 ABLATION ANALYSIS

An ablation analysis is performed and reported quantitatively in Tab. 7, and visually in appendix I for the task of facial hair content transfer. Without $\mathcal{L}_{Recon1}^{B}$ and $\mathcal{L}_{Recon1}^{A}$, the masks produced are

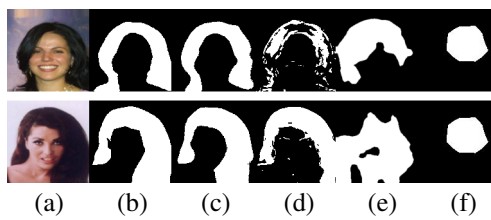

| (a) | (b) | (c) | (d) | (e) | (f) |

Figure 8: Segmentation of women's hair. (a) original image, (b) ground truth segmentation, (c) our results, (d) the results of Press et al. (2019), (e) the results of Ahn & Kwak (2018), (f) the results of CAM.

Table 7: The effect of removing losses. Shown are classifier accuracy, cosine similarity, KID, and percentage of mask from the total size of the face.

|  | Class. | Sim. | KID | Perc. |
|---|---|---|---|---|
| $\mathcal{L}$ | 88.1% | 0.89 | $3.5 \pm 0.1$ | 23% |
| w/o $\mathcal{L}_{Recon2}^{A}$ | 88.5% | 0.85 | $4.1 \pm 0.5$ | 34% |
| w/o $\mathcal{L}_{Recon2}^{B}$ | 88.1% | 0.87 | $4.2 \pm 0.4$ | 65% |
| w/o $\mathcal{L}_{Cycle}$ | 67.1% | 0.95 | $4.1 \pm 0.9$ | 29% |
| w/o $\mathcal{L}_{Recon1}^{B}$ | 9.4% | 1.0 | $4.3 \pm 0.7$ | 0% |
| w/o $\mathcal{L}_{Recon1}^{A}$ | 9.7% | 1.0 | $4.6 \pm 1.0$ | 0% |
| w/o $\mathcal{L}_{DC}$ | 9.5% | 1.0 | $5.0 \pm 1.0$ | 0% |
| L2 reg | 88.0% | 0.82 | $4.6 \pm 0.7$ | 33% |
| L2 recon #1 | 87.7% | 0.89 | $3.3 \pm 0.5$ | 22% |
| L2 recon #2 | 74.2% | 0.93 | $4.2 \pm 0.6$ | 30% |

empty and hence facial hair is not transferred to the target image, indicated by the high cosine similarity values but low classifier scores (i.e., the classifier labels the output as belonging to domain $A$). Similarly, without $\mathcal{L}_{DC}$, masks produced are empty as no disentanglement is possible.

Without $\mathcal{L}_{Cycle}$ the masks produced include larger portions of the face, which also maintains similarity but hurts the classification score. $\mathcal{L}_{Recon2}^{B}$ and $\mathcal{L}_{Recon2}^{A}$ play a lesser role for the mask. Without $\mathcal{L}_{Recon2}^{B}$ the mask is less smooth, and without $\mathcal{L}_{Recon2}^{A}$ the mask still captures additional objects (e.g eyes). In fact, $\mathcal{L}_{Recon2}^{A}$ is a way to enforce the mask to capture the relevant content in a self-regularizing way. $\mathcal{L}_{Recon2}^{A}$ and $\mathcal{L}_{Recon2}^{B}$ are dependent on z, which is semantically aware of the domain specific content, while $L2$ equally penalizes any region of the image regardless of its content. When trying to use L2 norm, the mask had to be carefully adjusted to each experiment and resulted in a non-smooth mask which covers unnecessary parts of the face. This can be seen visually in appendix Fig 36 and from the "L2 reg" entry of Tab. 7, where L2 regularization is used instead of $\mathcal{L}_{Recon2}^{A}$ and $\mathcal{L}_{Recon2}^{B}$. We note that Chen et al. (2018) uses sparsity regularization on the masks and Mejjati et al. (2018) uses early stopping, which we do not require due to the regularization of $\mathcal{L}_{Recon2}^{A}$ and $\mathcal{L}_{Recon2}^{B}$. Refer to appendix H for further discussion.

## 5    CONCLUSIONS

When transferring content between two images, we need to know what to transfer, where to transfer it to, and how to transfer it. Previous work in guided transfer either transferred global style properties or neglected the "where" aspect, which ultimately lead to an ineffective generation that lacks attention.

In our work, the "what" aspect is captured by $E_s$, and $D_B$ captures both the "where" and the "how". Our results demonstrate that the context (image $a$) in which the content is placed determines not just the location of the inserted content but also the form in which it is presented, where both aspects can vary dramatically, even for a fixed content-guide image $b$. The comprehensive modeling of the guided content transfer problem leads to results that are far superior to the current state of the art. In addition, the modelling of "where" allows us to obtain accurate segmentation masks in a weakly supervised way, remove content, swap content between images, and add multiple contents without suffering a gradual degradation in quality.

## ACKNOWLEDGEMENTS

This project has received funding from the European Research Council (ERC) under the European Union's Horizon 2020 research and innovation programme (grant ERC CoG 725974).

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

## A  ARCHITECTURE AND HYPERPARAMETERS

We consider samples in $A$ and $B$ to be images in $\mathbb{R}^{3 \times 128 \times 128}$. The encoders $E_c$ and $E_s$ each consist of 6 convolutional blocks. Similarly, $D_A$ and $D_B$ consist of 6 de-convolutional blocks.

A convolutional block $d_k$ consisting of: (a) $4 \times 4$ convolutional layer with stride 2, pad 1 and $k$ filters (b) a batch normalization layer (c) a Leaky ReLU activation with slope 0.2. Similarly, a de-convolutional block $u_k$ consists of: (a) $4 \times 4$ de-convolutional layer with stride 2, pad 1 and $k$ filters (b) a batch normalization layer (c) a ReLU activation.

The structure of the encoders and decoders is then: $E_c$: $d_{32}, d_{64}, d_{128}, d_{256}, d_{512-sep}, d_{512-2 \cdot sep}$, $E_s$: $d_{32}, d_{64}, d_{128}, d_{128}, d_{128}, d_{sep}$, $D_A$: $u_{512}, u_{256}, u_{128}, u_{64}, u_{32}, u_3^*$, and $D_B$: $u_{512}, u_{256}, u_{128}, u_{64}, u_{32}, u_4^*$.

The last layer of $D_A$ ($u_3^*$) differs in that it does not contain batch normalization and tanh activation is applied, instead of ReLU. $D_B$'s last layer ($u_4^*$) similarly does not contain batch normalization. The output is of size $4 \times 128 \times 128$. We split the output to a mask (first channel) and raw output (other three channels). We apply a sigmoid activation to the mask to get values between 0 and 1 and a tanh

activation for the raw output and a Tanh activation for the raw output. $sep$ is the dimension of the separate encoders, set to be 100 for all datasets.

The discriminator $C$ consists of a fully connected layer of 512 filters, a Leaky ReLU activation with slope 0.2, a second fully connected layer of one filter and a final sigmoid activation.

We use the Adam optimizer with $\beta_1 = 0.5, \beta_2 = 0.999$, and learning rate of 0.0002. We use a batch size of size 32 in training.

We constructed the train/test sets using 90%-95% split. This consists of about 7,200-18,000 examples for train and about 800-2,000 examples for test for each attribute.

## B    ADDITIONAL CONTENT TRANSFER RESULTS

Additional results to the ones presented in the main text are provided here.

Fig. 9 gives a comparison of our method to the state-of-the-art for the transfer of facial hair. Fig. 10 provides additional interpolation results for this task while Fig. 11 provides additional content transfer results. Fig. 12 shows the masks generated for this content transfer. Fig. 13 gives an example of the raw output given by our method for this task.

Fig. 14 gives additional results for the task of glasses transfer, while Fig. 15 shows the masks generated for this content transfer. Fig. 16 provides additional comparison to the baseline method.

Fig. 17 and Fig. 18 provide additional content transfer results and the generated masks for the task of smile transfer. It is well known that smile includes not only the mouth but also other facial features such as eyebrows and cheeks Korb et al. (2014), thus when our method transfer the smile, it transfer all the relevant facial features for the smile as can be seen in the generated masks in Fig. 18. Fig. 19 provide interpolation results and Fig. 20 gives a comparison of our method for this task.

Fig 21 gives a comparison of our method to Mejjati et al. (2018). While our method uses guidance image which allows one to many translation, Mejjati et al. (2018) can translate to only one image.

## C    ADDITIONAL OUT OF DOMAIN MANIPULATIONS RESULTS

Fig. 24 shows sample results where the mapping of facial hair is applied to female faces. Out of distribution translation where the train domains are different from the inference domains are shown for glasses in Fig 23. We further consider the ability of our method to perform translation on images from the out-of-distribution LFW dataset Huang et al. (2007) (Fig.22(a)), as well as images of an alien and a baby (Fig.22(b)) which are extremely out of distribution, all not present during training. As can be seen, even in these cases, our method successfully transfers the desired content. .

## D    COMPARATIVE RESULTS FOR THE HANDBAG DATASET

Fig. 25 gives a comparison of our method for the handle transfer for handbags, while Fig. 26 provides more results for this task (on top of Fig. 5).

## E    A QUALITATIVE COMPARISON OF OUR METHOD TO LITERATURE METHODS ON THE ATTRIBUTE REMOVAL TASK

Fig. 27 gives a comparison of our method on the task of attribute removal. The translation of Press et al. (2019) is blurry and suffers from many of the facial features being lost. For example, for glasses removal, the men on the right have facial hair which is lost in the translation. Lample et al. (2017) completely changes the facial features. For example, for facial hair removal, the gender seems to change from men to women. He et al. (2019) and Liu et al. (2019) are unable to remove the mustache and the translation is of lower quality in general. For example, for the glasses removal, for the man in the middle, the translation is unnatural around the eyes. Our translation is of consistently higher quality for all tasks and successfully removes the desired attribute.

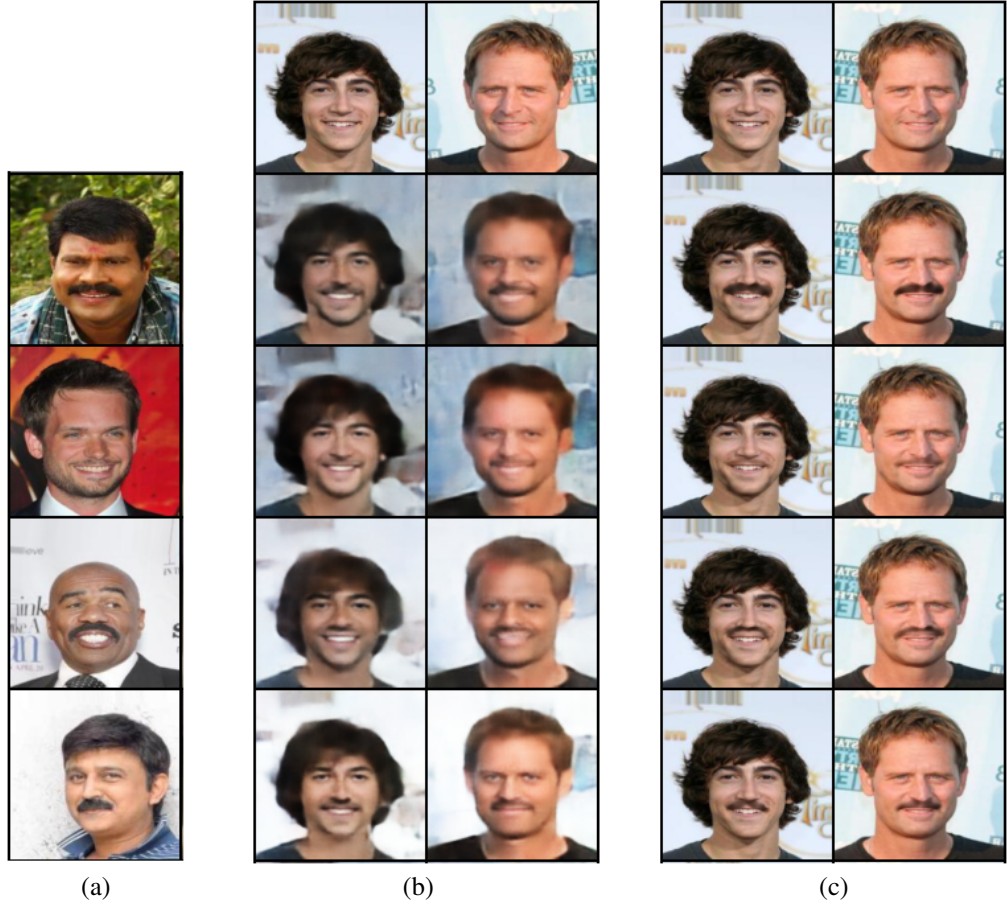

(a)                            (b)                            (c)

Figure 9: (a) Guide images in domain $B$ (faces with facial hair). (b) Results by the method of Press et al. (2019): the top row is the source images in domain $A$. The others incorporate the facial hair from the corresponding row of (a). (c) Same mapping for our method.

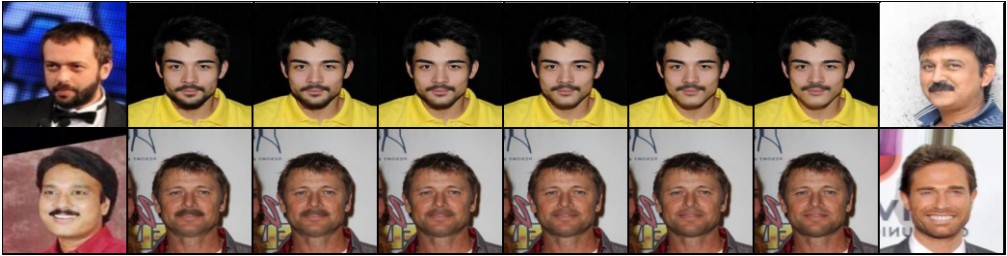

Figure 10: Facial Hair Interpolation

# F    ADDITIONAL SEQUENTIAL CONTENT TRANSFER AND ATTRIBUTE REMOVAL RESULTS

We provide additional images produced by our method as well as by the baseline method. As can be seen, in order to perform the guided content transfer of two attributes from two different domain, Press et al. (2019) passes the source input image into the network twice which wastefully reconstructs static facial features twice.

For example, for sequential addition of glasses and facial hair, as seen in Fig 28, our method successfully transfers the two attributes, while Press et al. (2019) not only produces blurry images,

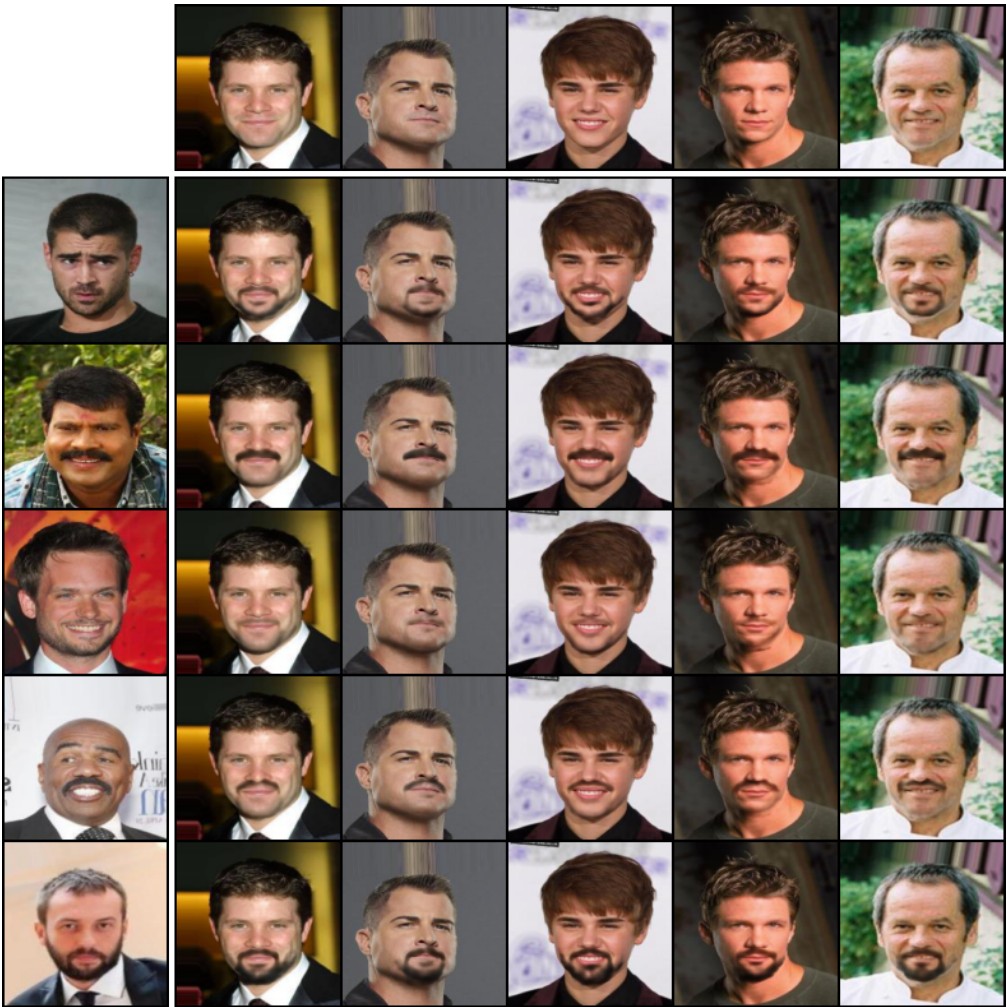

Figure 11: Additional results for the guided transfer of facial hair.

but is much worse at transferring the content from both attributes. This observation is also supported by the user study performed in Tab 4.

Fig. 29 show the comparison to the baseline model for the task of closing the mouth and adding glasses; Fig. 30 shows additional results from our method for this task. Fig. 31 provides the comparison for the task of replacing facial hair; while Fig. 32 presents additional results for this task.

## G   ADDITIONAL WEAKLY SUPERVISED SEGMENTATION RESULTS

Fig 33 gives a comparison our method for the task of men's hair segmentation as given in section 4.2 of the main text, while Fig 34 gives additional results for the segmentation of woman's hair.

Additional segmentation results are shown in Fig. 35 for the domain of glasses and facial hair. In this domain quantitative results cannot be obtained due to lack of ground truth segmentations.

## H   ADDITIONAL ABLATION STUDY DISCUSSION

While the loss in Eq. 7 directly affects the mask generation, the losses in Eq. 3 ($\mathcal{L}_{DC}$) and Eq. 5 ($\mathcal{L}_{Recon1}^{A}$) affect it indirectly. Without the loss in Eq. 3 ($\mathcal{L}_{DC}$), no disentanglement is possible, and the common encoder would contain all of the image information including the separate information.

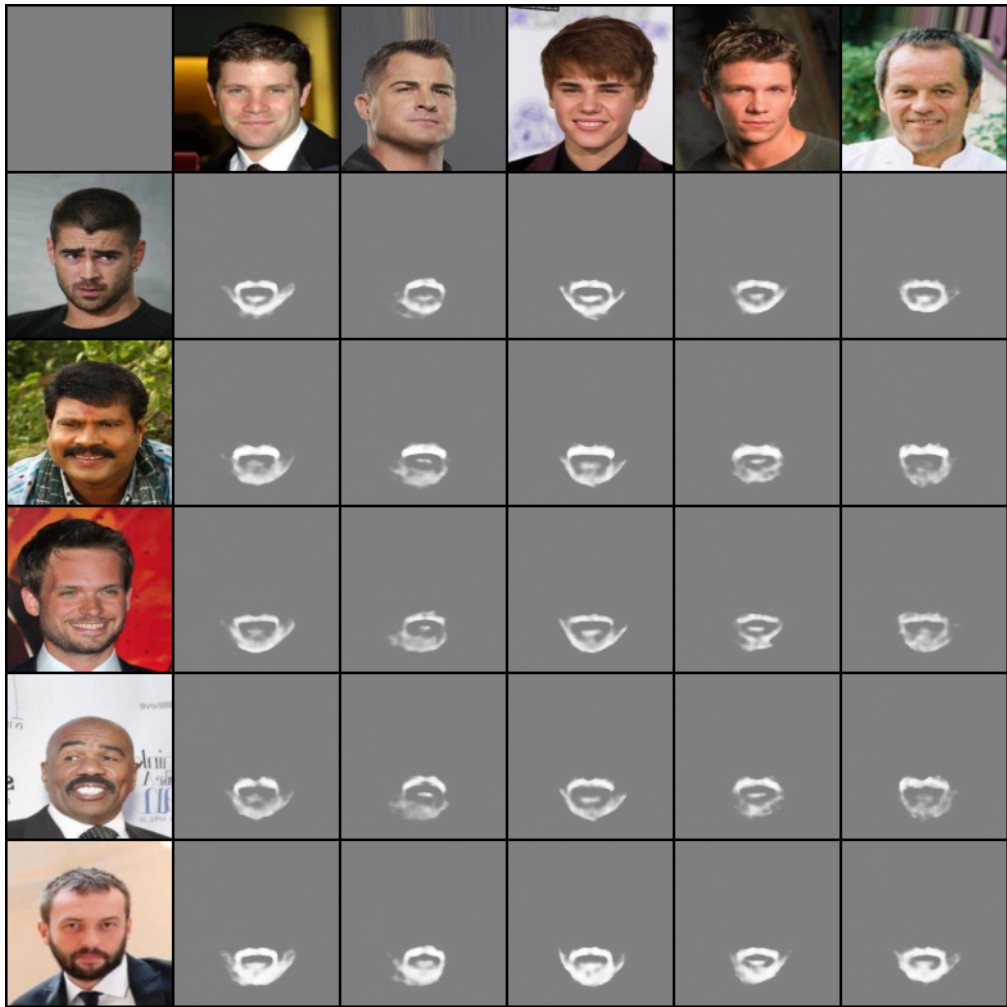

Figure 12: Masks generated for the guided transfer of facial hair experiment. Masks generated are for the translated images in Fig 11.

This means that the image produced by $D_A(E_c(b))$ is close to $b$ and, therefore, the generated mask is empty.

Furthermore, without the loss of Eq. 5 ($\mathcal{L}_{Recon1}^A$), we empirically observe that $D_A(E_c(b))$ outputs the image with the specific part intact (for example, the facial hair is not removed). This indirect effect on the disentanglement probably stems from the fact that without this loss, there is reconstruction only on faces with facial hair (running example for the specific part). Thus, $E_c$ can encode generic facial hair information for shaved faces and have $E_c(b)$ and $E_c(a)$ still indistinguishable. Eq. 5 ($\mathcal{L}_{Recon1}^A$) makes sure that $E_c$ won't encode facial hair for shaved faces, since it requires reconstruction of an image without facial hair.

We further consider the effect of replacing the norm used for reconstruction losses from L1 to L2. When using L2 norm for $\mathcal{L}_{Recon1}^B$ and $\mathcal{L}_{Recon1}^A$ the result is comparable both numerically ("L2 recon #1" in Tab. 7) and visually (see appendix Fig 36). When using L2 norm for $\mathcal{L}_{Recon2}^B$ and $\mathcal{L}_{Recon2}^A$ ("L2 recon #2" in Tab. 7) the results change more significantly: the size of the mask is larger and the classifier score significantly lower. We attribute this to the sparsity inducing effect of the L1

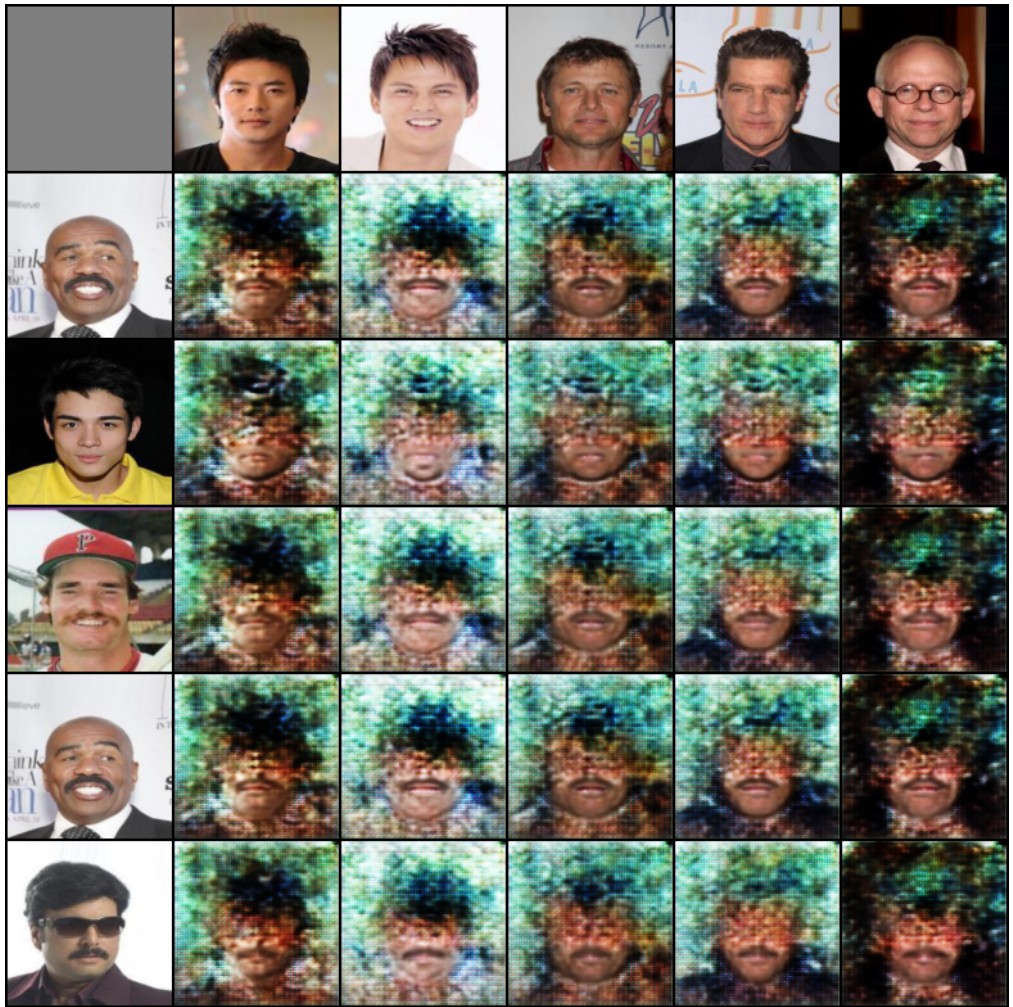

Figure 13: Raw outputs generated by $D_b$ for the task of facial hair content transfer.

## I    VISUAL RESULTS OF THE ABLATION STUDY

Fig 36 shows the masks generated when different losses are removed as discussed in the ablation analysis of section 4.3 of the main text.

## J    HYPERPARAMETERS SENSITIVITY

The $\lambda$ coefficients were set in a way that reflects their relative importance and were observed. For example, if the mask obtained was too large we would increase $L_{Recon2}^{A}$ and $L_{Recon2}^{B}$. As illustrated in Fig. 38 and Fig. 39, our network is not overly sensitive to the choice of these values. For example, for $L_{Recon2}^{A}$, each value in the range 0.4-1.0 results in a similar output and for $L_{Recon1}^{A}$ each value in the range 3.0-7.0 results in a similar output.

Fig. 37 shows the affect of the threshold used to binarized the mask for the task of segmentation. Fig. 38 and Fig. 39 show that the network is not overly sensitive to the choice of $\lambda_4$ and $\lambda_1$ respectively.

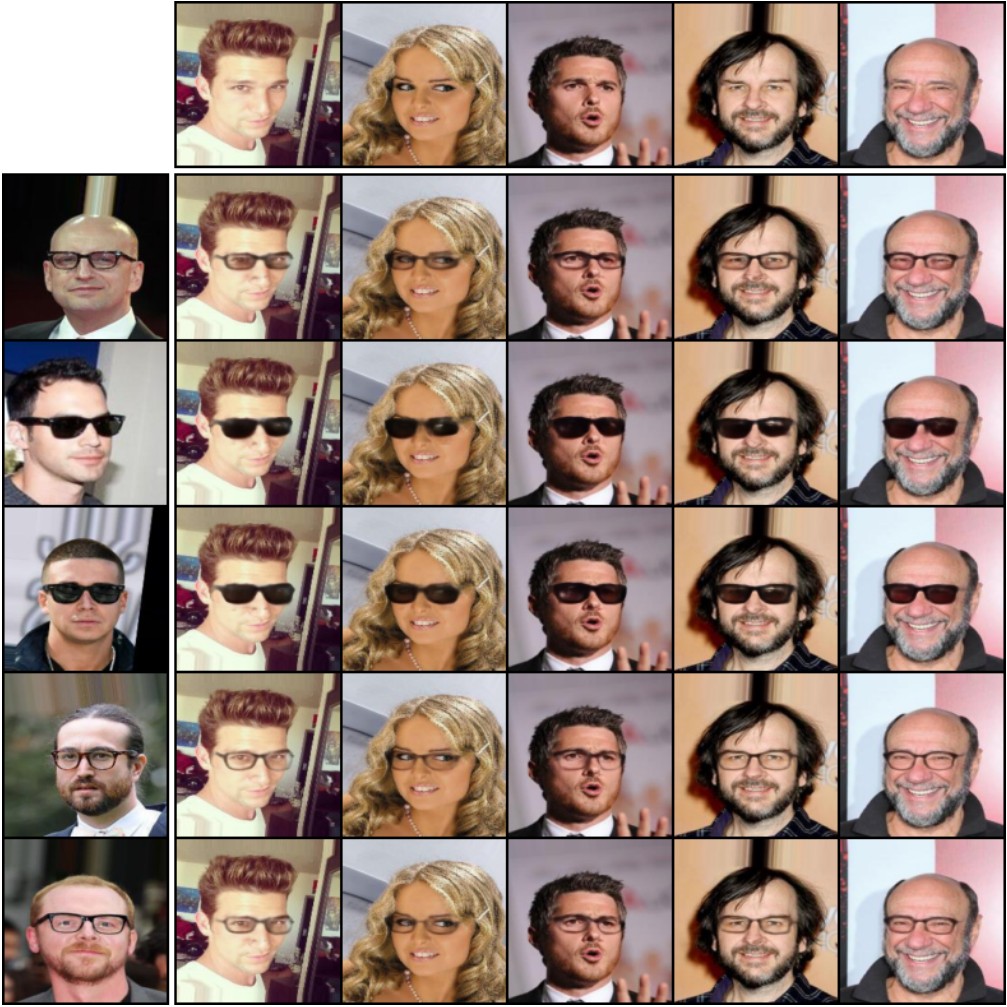

Figure 14: Additional content transfer example. Given an image with glasses (left), and another image of a face with no glasses (top), the proposed method identifies and translates the specified glasses from the former domain to the latter.

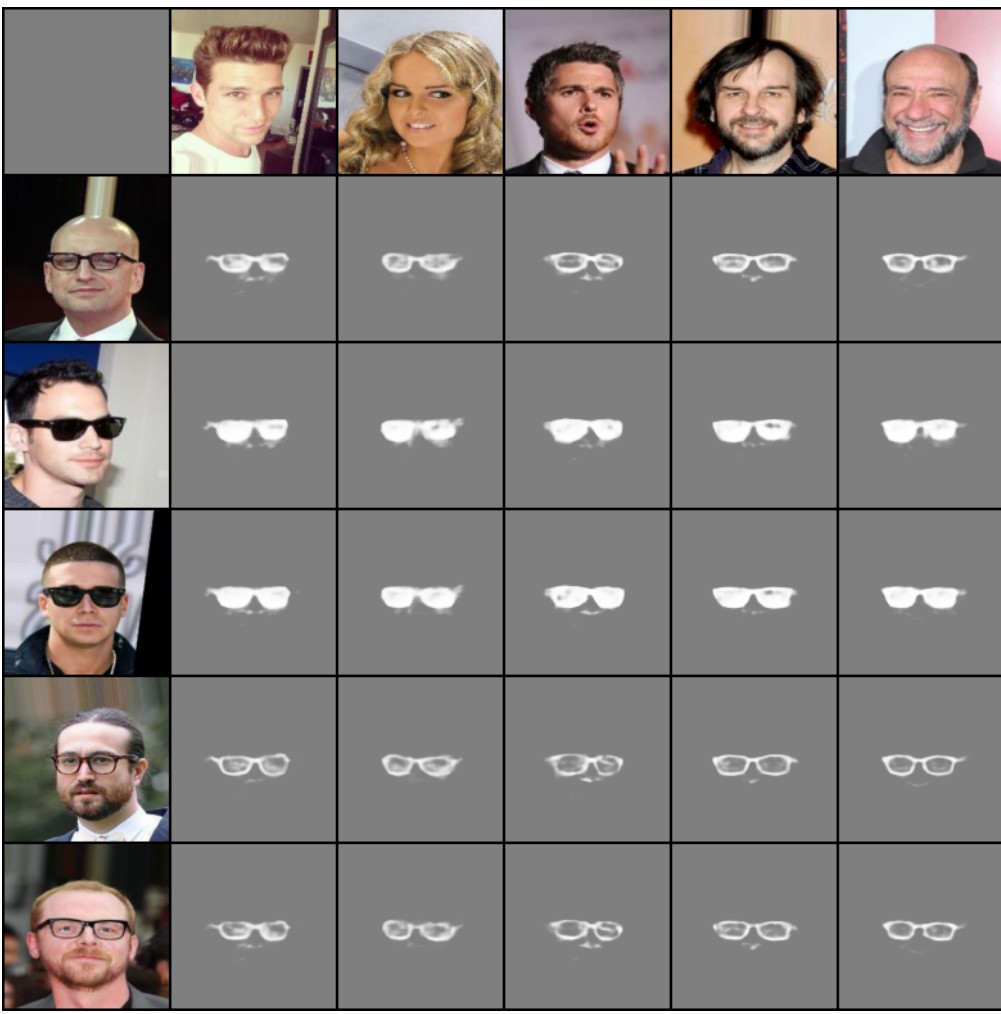

Figure 15: Masks generated for the guided transfer of glasses experiment. Masks generated are for the translated images in Fig 14.

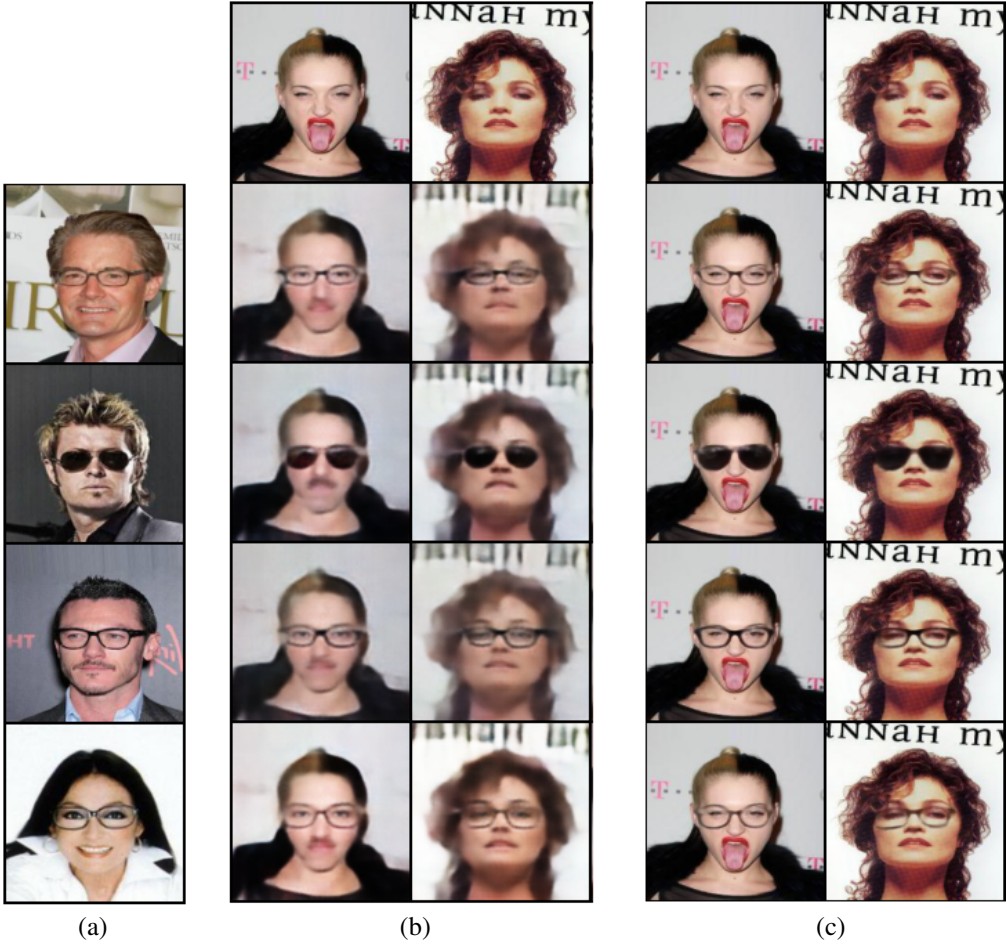

(a)               (b)              (c)

Figure 16: (a) Guide images in domain $B$ (glasses). (b) Press et al. (2019) method: the top row is the source images in domain $A$. The others incorporate the glasses from the corresponding row of (a). (c) Same mapping for our method.

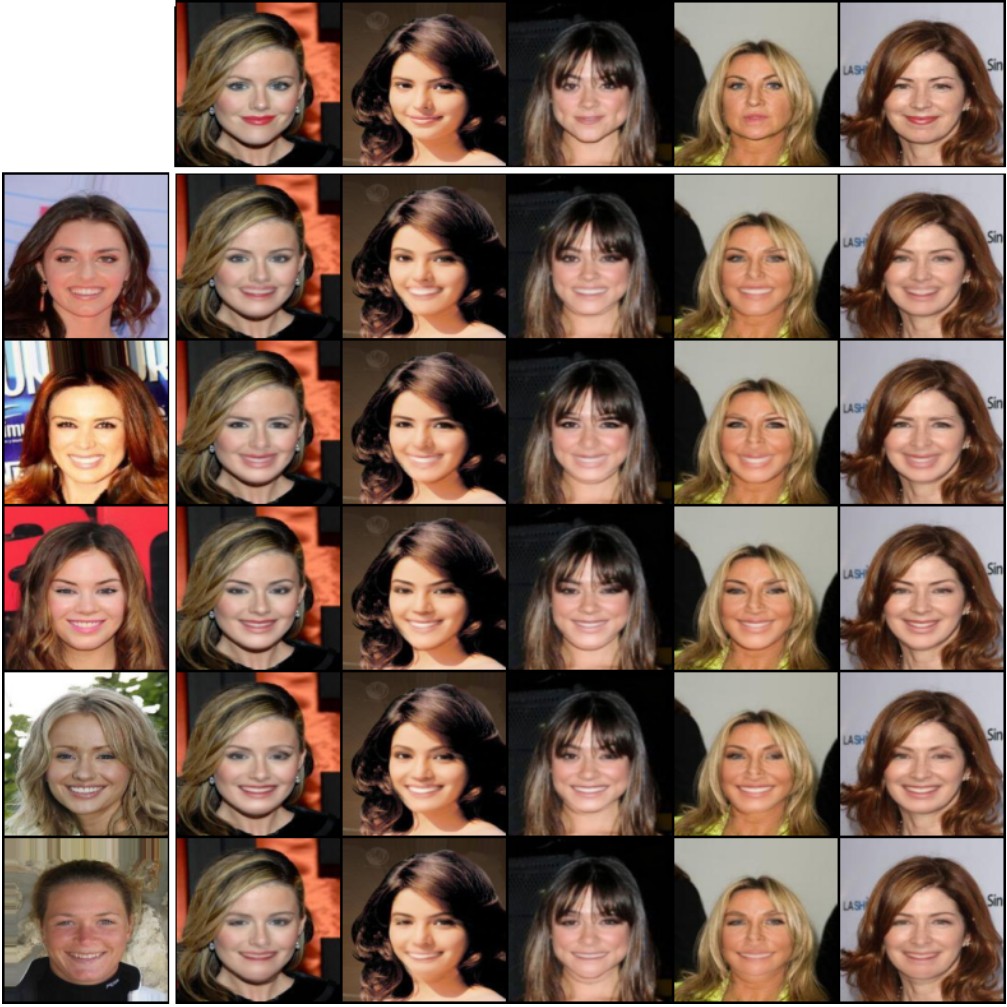

Figure 17: Additional content transfer example. Given an image of smiling face (left), and another image of a non-smiling face (top), the proposed method identifies and translates the specified smiles from the former domain to the latter.

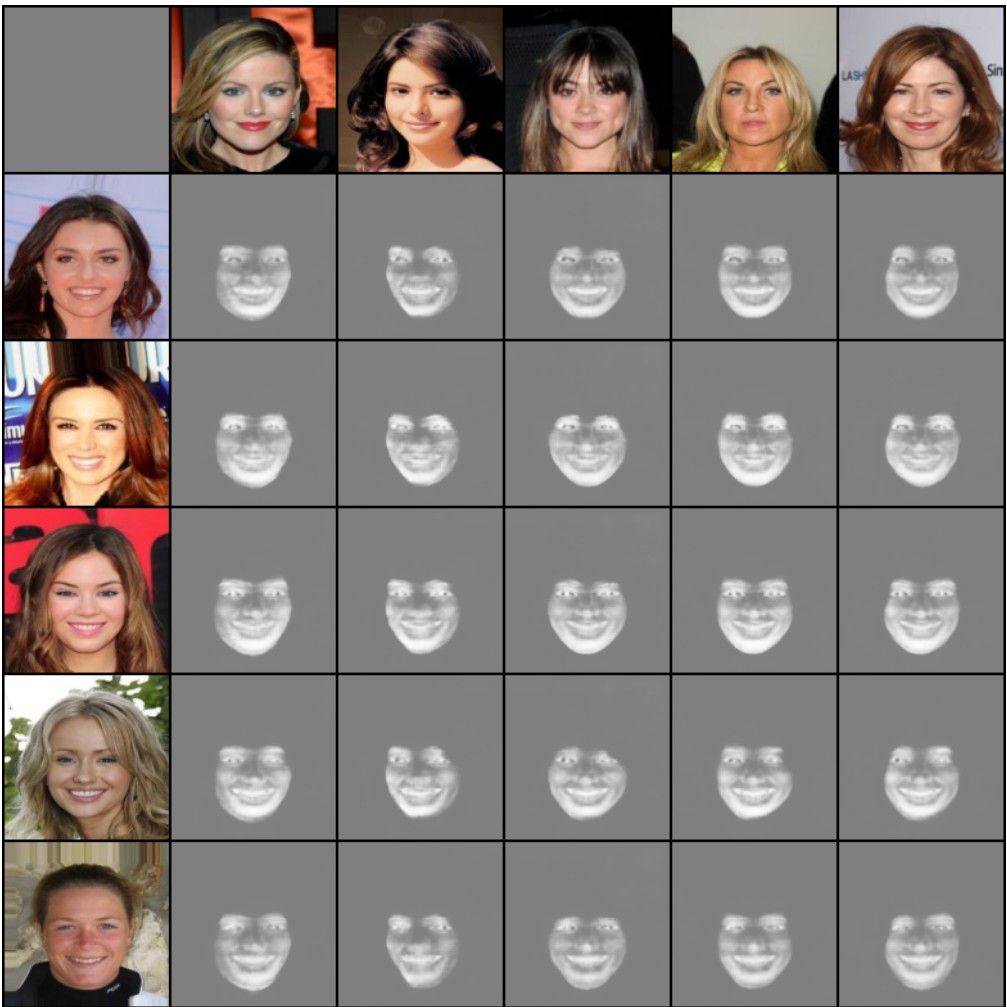

Figure 18: Masks generated for the guided transfer of smile experiment. Masks generated are for the translated images in Fig 17.

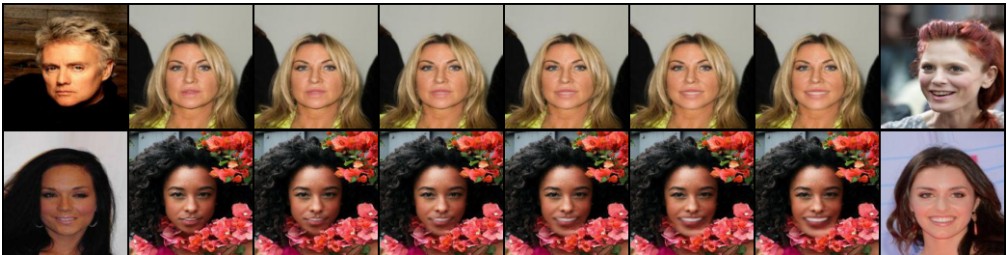

Figure 19: Smile Interpolation

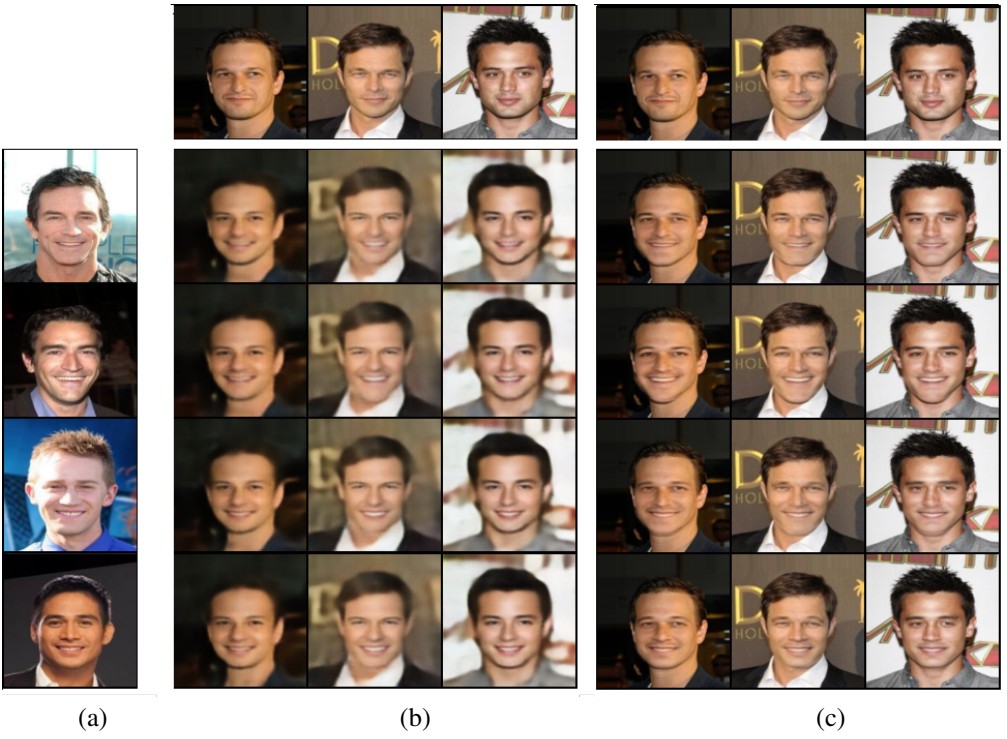

(a)                   (b)                   (c)

Figure 20: (a) Guide images in domain $B$ (faces with smile). (b) Press et al. (2019): the top row is the source images in domain $A$. The others incorporate the smile from the corresponding row of (a). (c) Same mapping for our method.

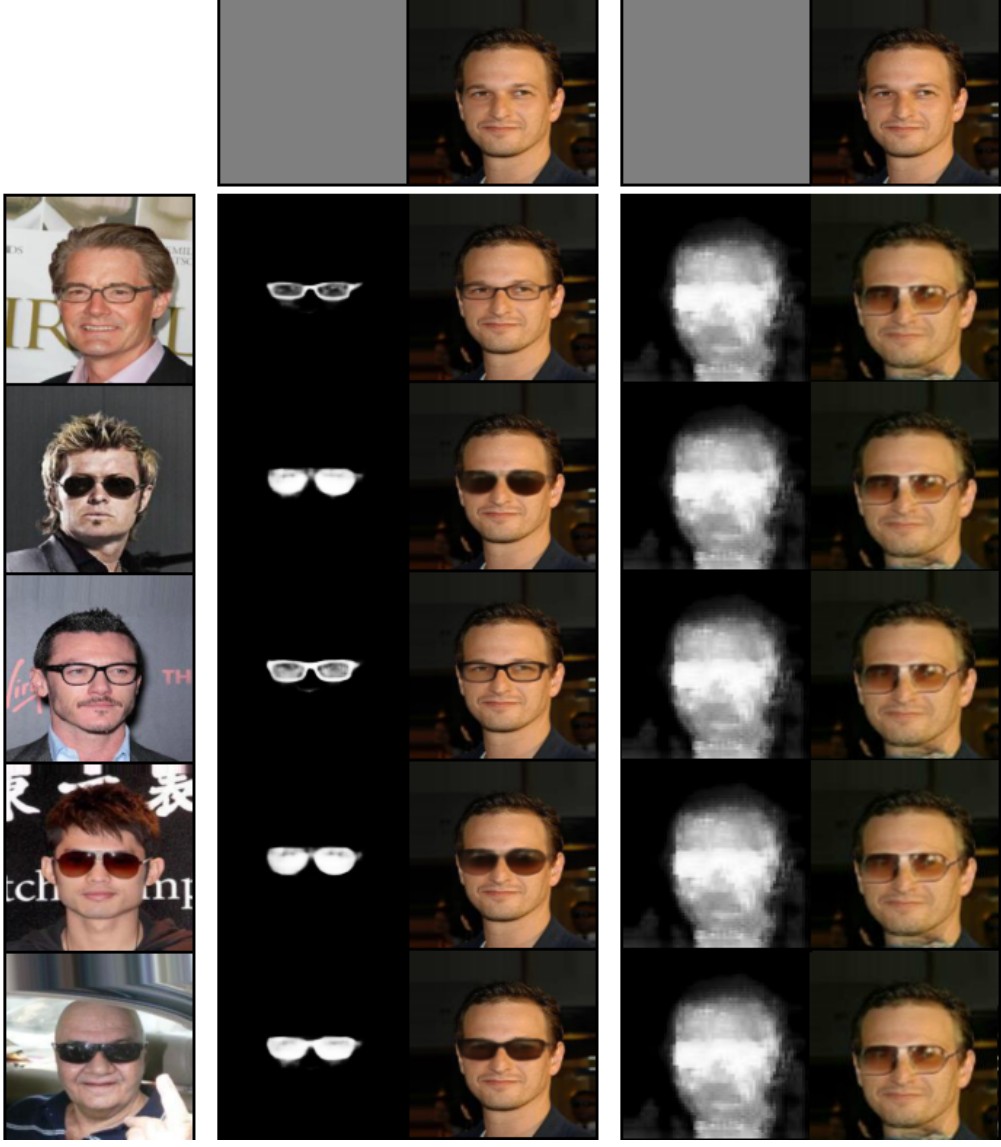

Figure 21: Our method (left) compared to Mejjati et al. (2018) (right) on the task of adding glasses to the original image (top). We show the generated masks and the final result for both methods. For our method we also show the guidance images.

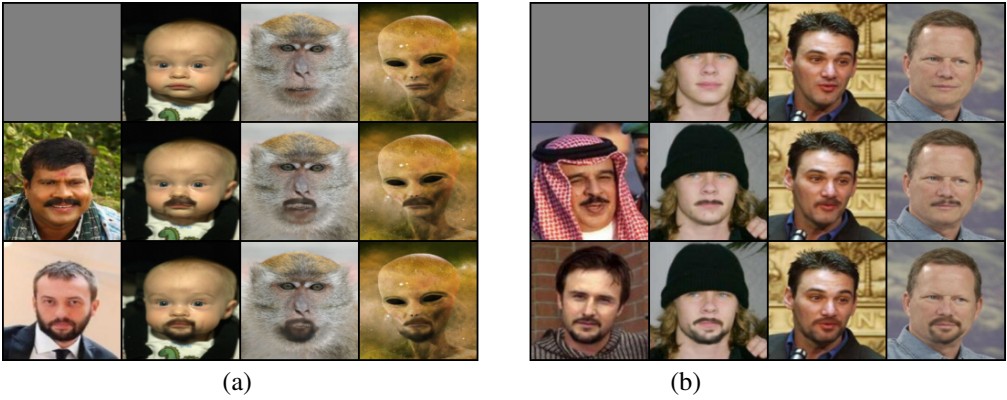

Figure 22: Out of domain translation. (a) Results on extremely out of domain images. (b) Results obtained by manipulating LFW images.

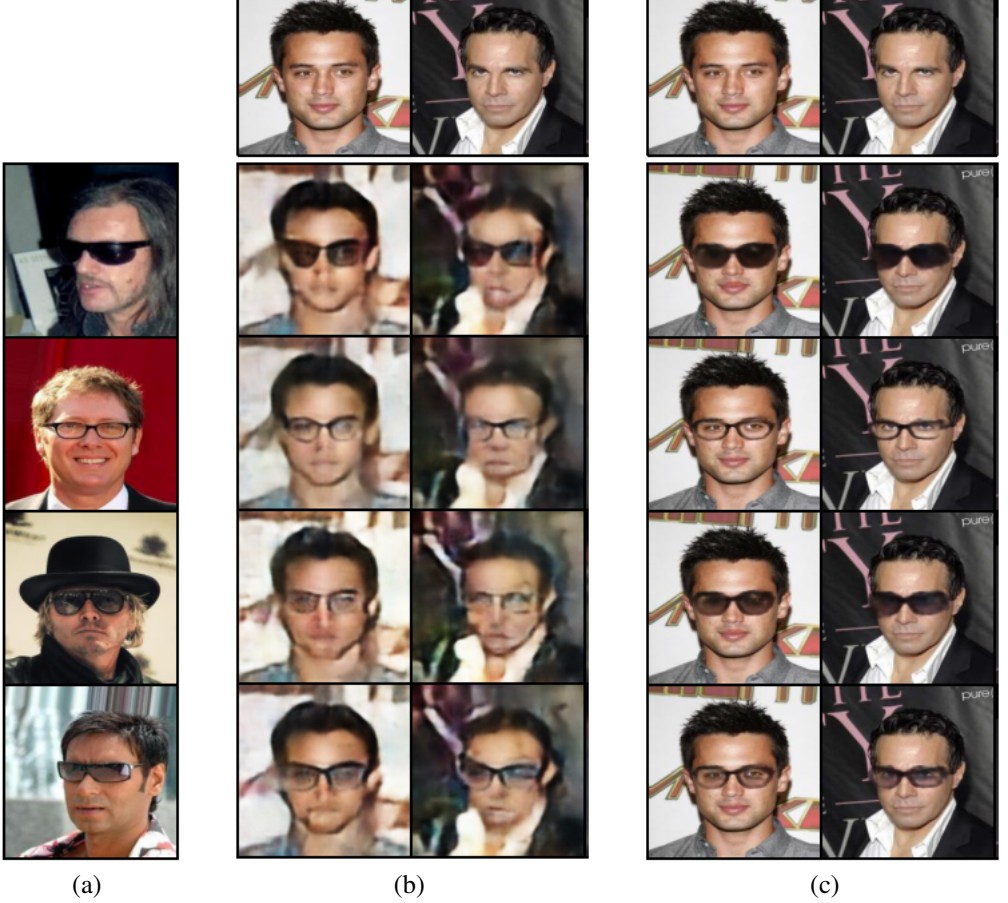

Figure 23: Out of distribution translation. The mapping between faces without and with glasses is trained only on women and applied to men. (a) Guide images in domain $B'$ (men with glasses) and top row is the source images in domain $A'$ (men without glasses), both not given during training. (b) The remaining rows are translated images by the method of Press et al. (2019). (c) Same mapping for our results.

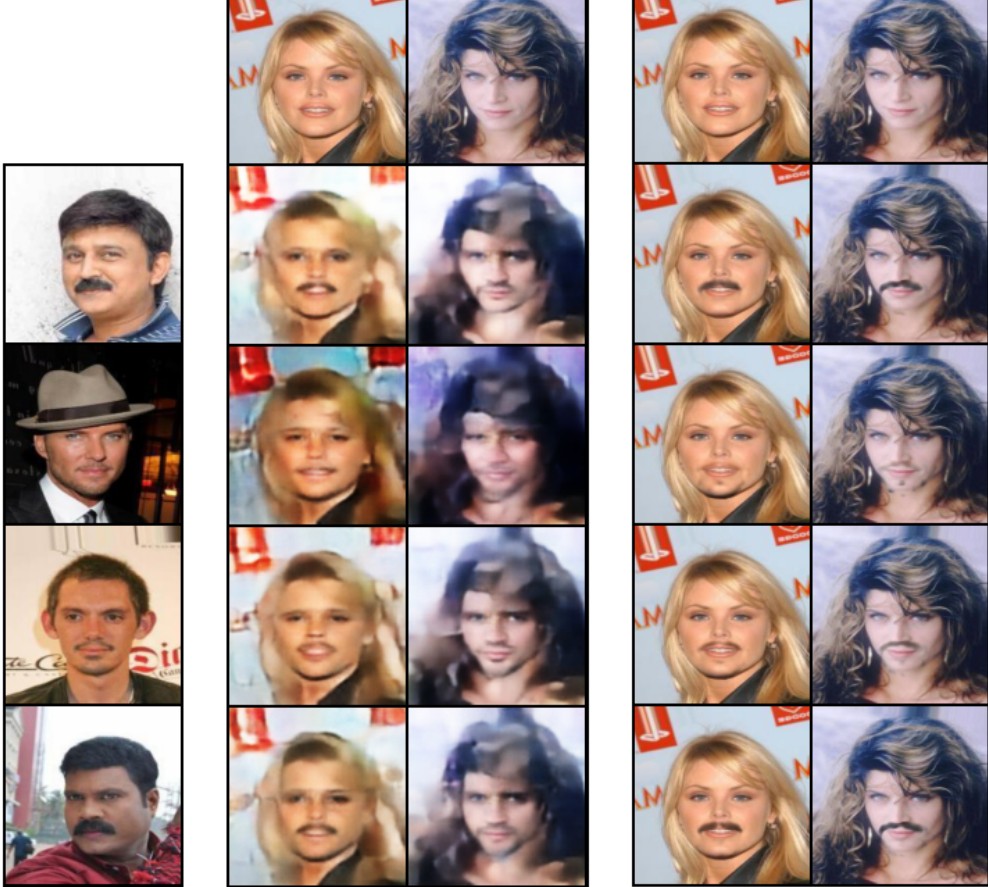

Figure 24: Additional out of distribution translation results. We train on mapping facial hair from male faces (left) to male faces, and apply this to women's faces (top) during inference time. Our domain translation results (right) are compared to those of Press et al. (2019) (middle).

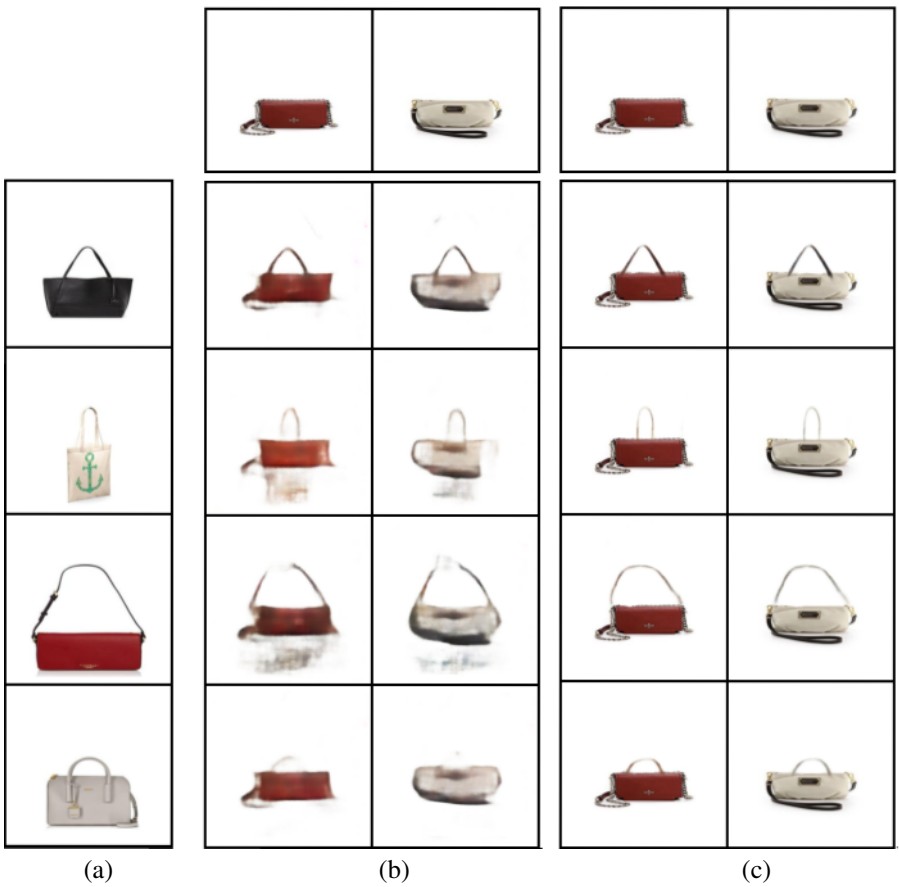

(a)                    (b)                    (c)

Figure 25: (a) Guide images in domain $B$ (handbags with handles). (b) Press et al. (2019): the top row is the source images in domain $A$. The others incorporate the handles from the corresponding row of (a). (c) Same mapping for our method.

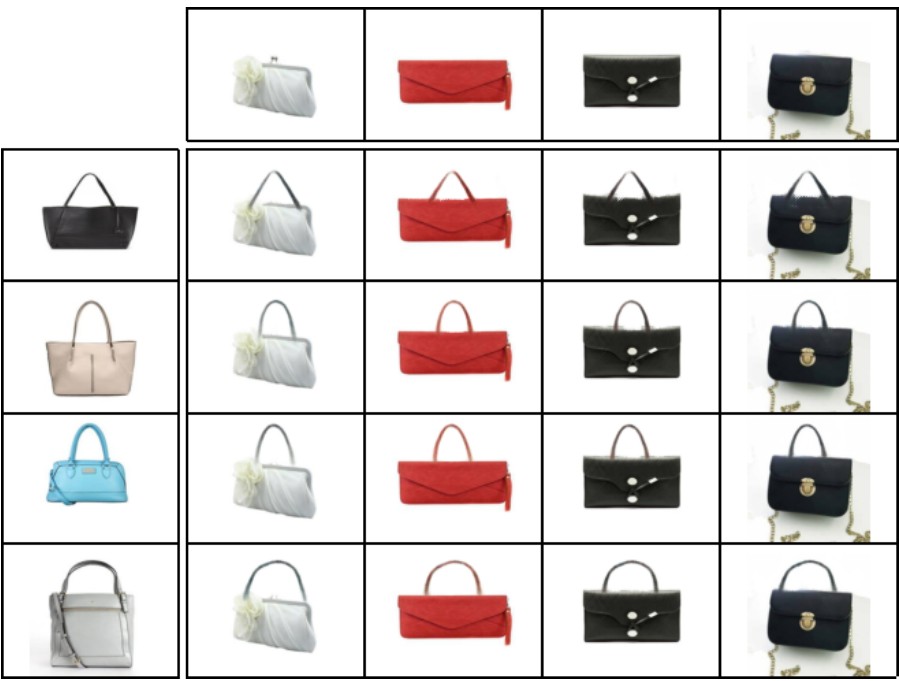

Figure 26: Additional content transfer example. Given an image of bag with a handle (left), and another image of a handbag (top), the proposed method identifies and translates the specified handbag from the former domain to the latter.

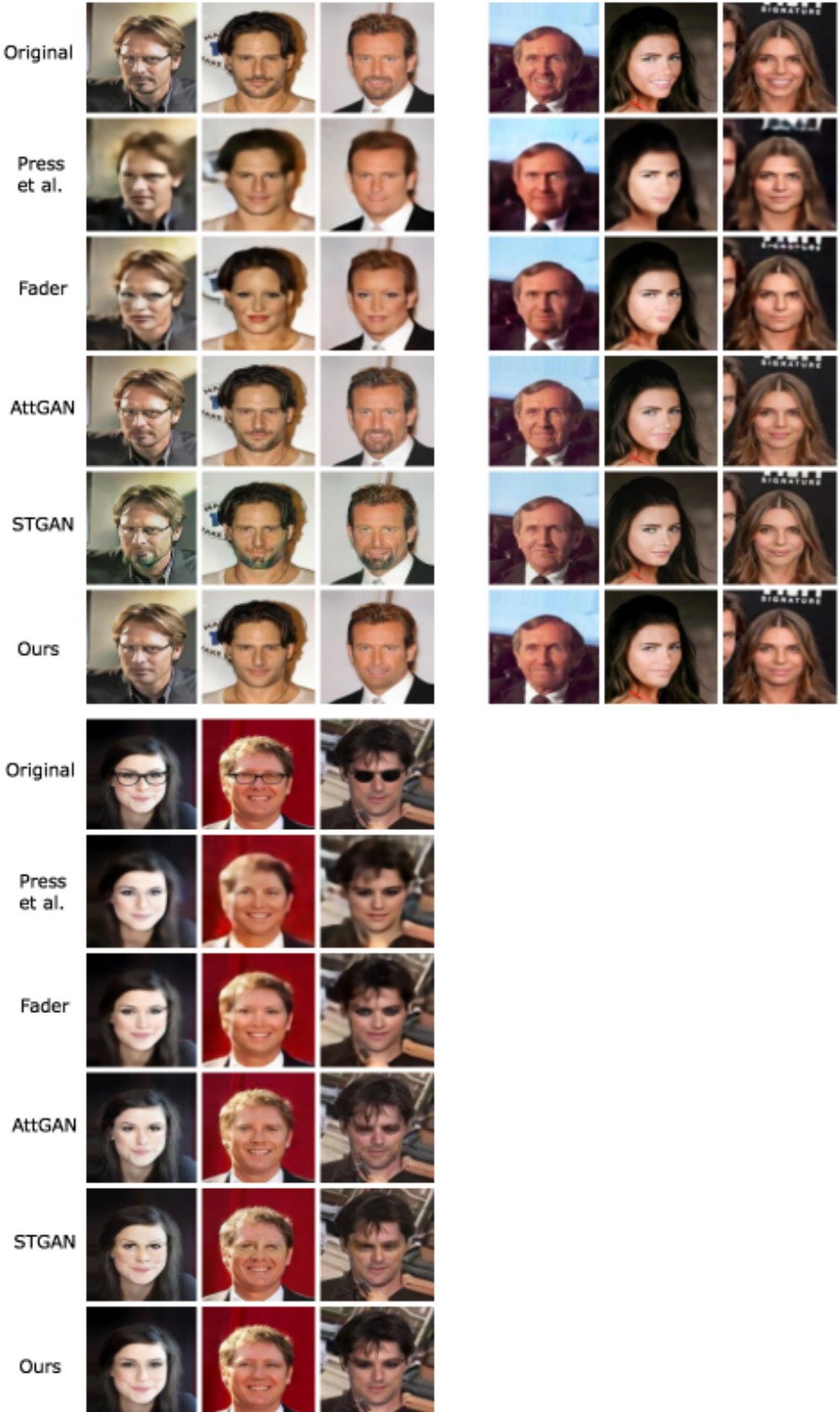

Figure 27: Attribute removal for the task of mustache (top left), smile (top right) and glasses (bottom left). The result of our method is shown alongside the baseline methods, Press et al. (2019), Fader (Lample et al. (2017)), AttGAN (He et al. (2019)) and STGAN (Liu et al. (2019)).

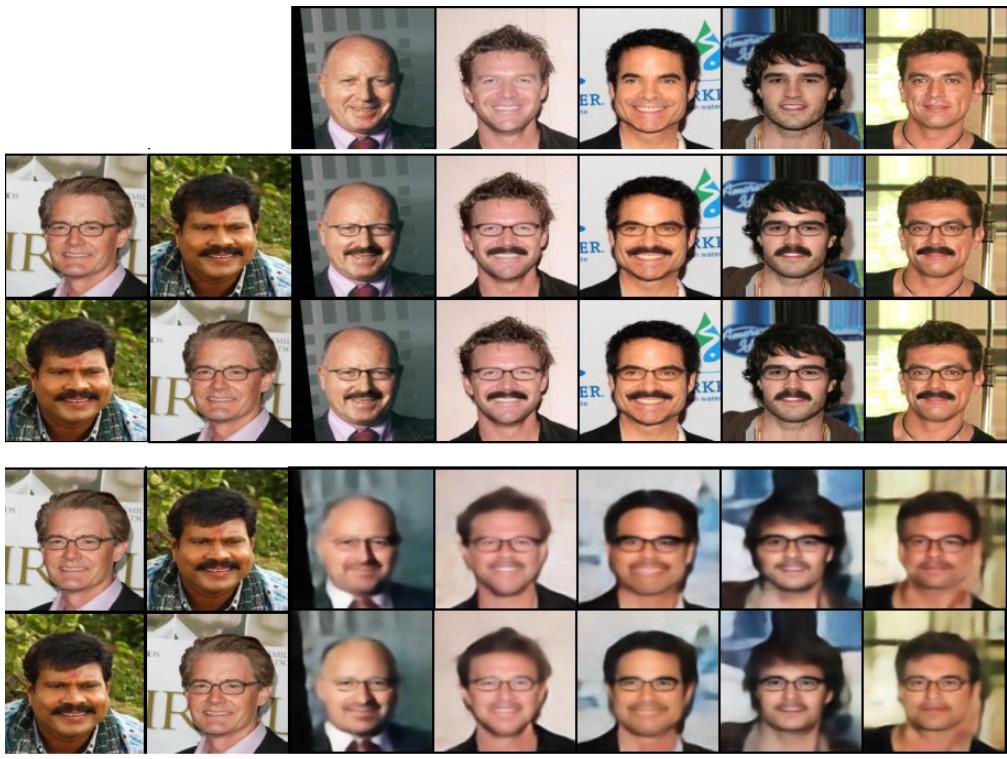

Figure 28: First two images on the left are the content donors applied sequentially, either with facial hair or glasses. The top row is the input source images. The results on the bottom are the translation of Press et al. (2019) while on the top are our results.

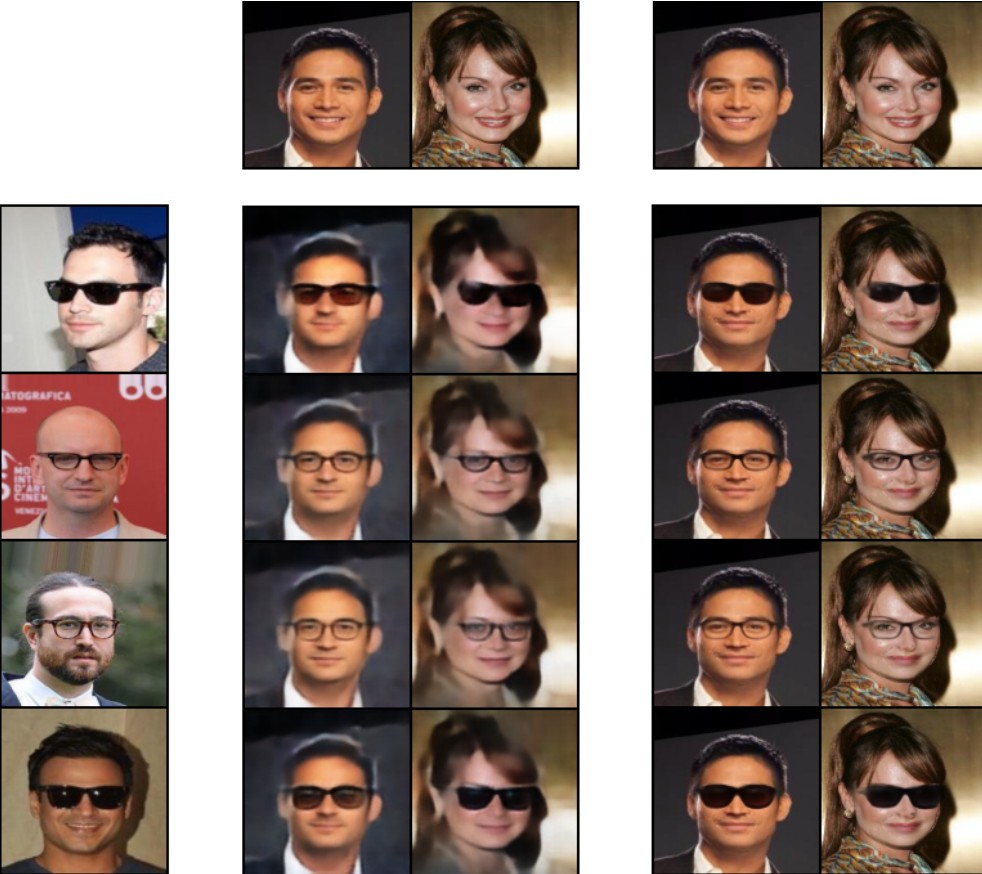

Figure 29: Additional removal and content transfer results. Given an image with glasses (left), and another image of a face with no glasses and a smile (top), the proposed method removes the smile and identifies and translates the specified glasses from the former domain to the latter. In the middle are the translated examples of Press et al. (2019) while on the right are our translated results.

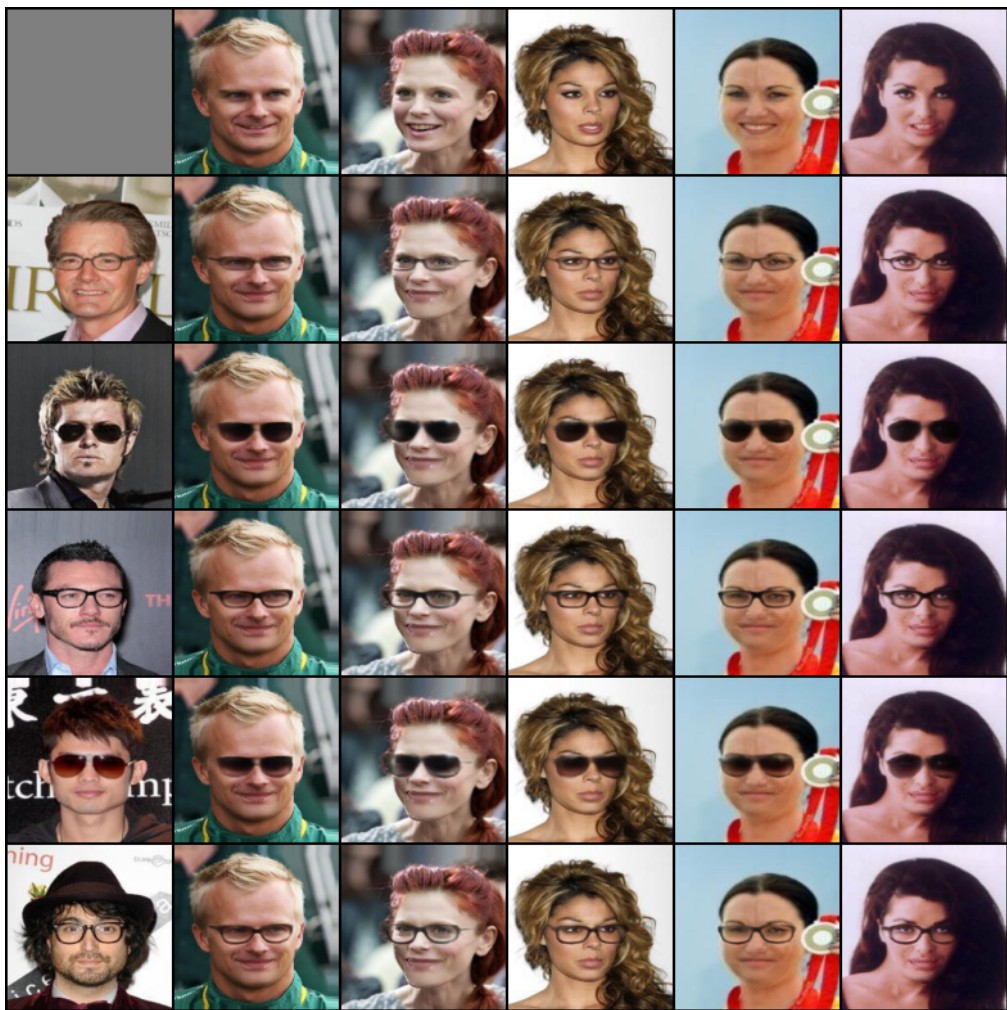

Figure 30: Additional removal and content transfer results for smile removal and glasses addition. Given an image with glasses (left), and another image of a face with no glasses and a smile (top), the proposed method removes the smile and translates the specified glasses from the former domain to the latter.

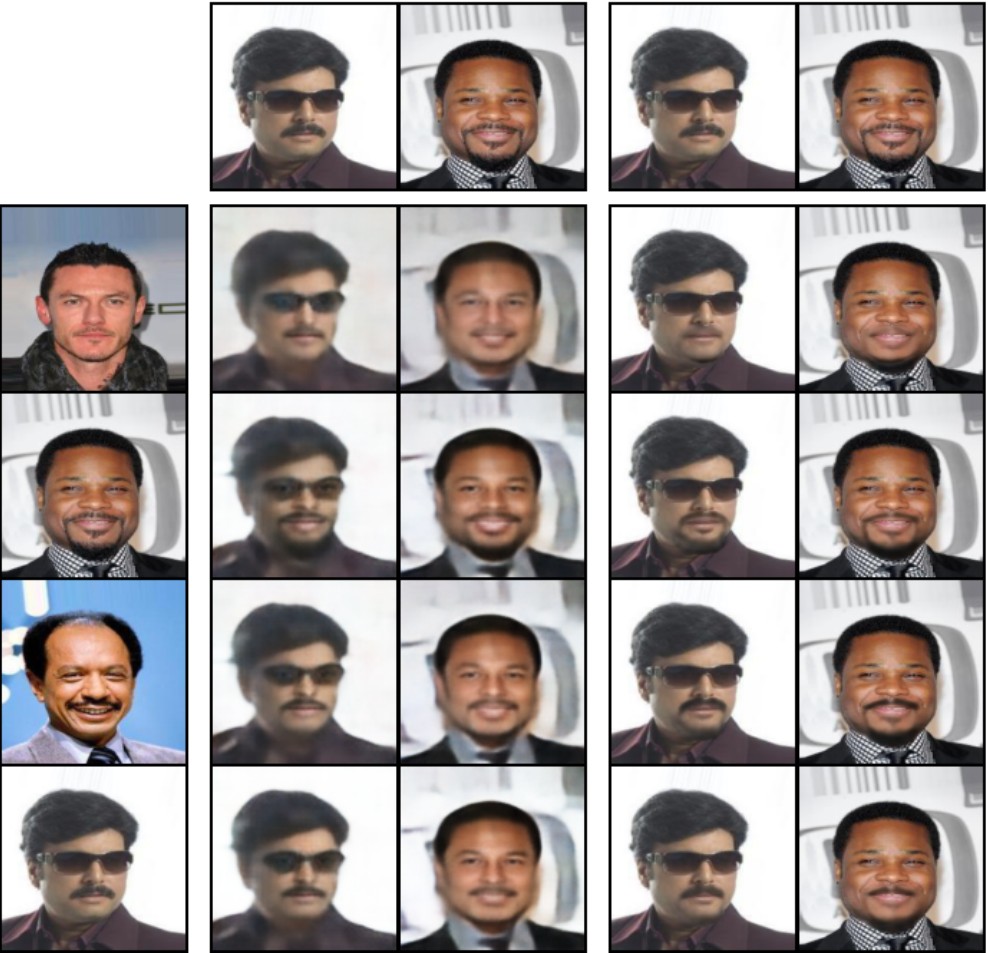

Figure 31: Facial hair swap results. Our method first removes the facial hair and then adds the facial hair of the guided image on the left. In the middle are the translated examples of Press et al. (2019) while on the right are our translated results.

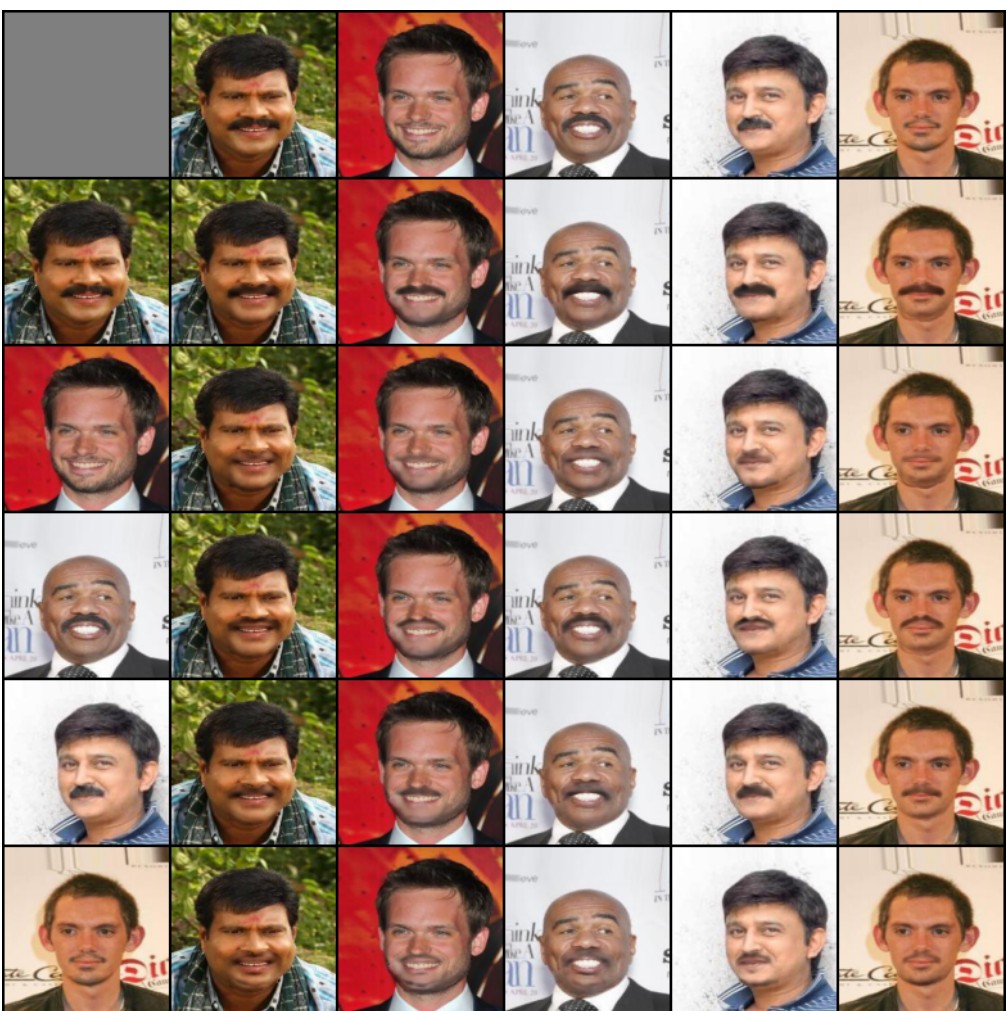

Figure 32: Additional facial hair swap results. Our method first removes the facial hair and then adds the facial hair of the guided image on the left

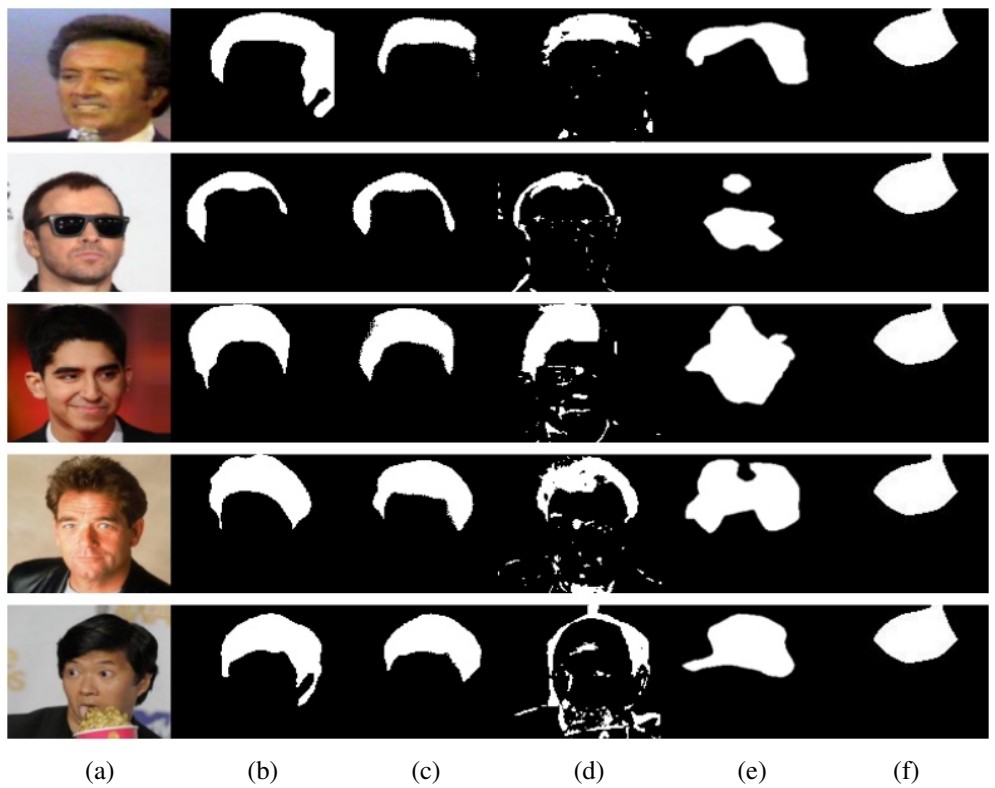

|  (a)  |  (b)  |  (c)  |  (d)  |  (e)  |  (f)  |

Figure 33: Segmentation of men's hair. (a) original image, (b) ground truth segmentation, (c) our results, (d) the results of Press et al. (2019), (e) the results of Ahn & Kwak (2018), (f) results of CAM.

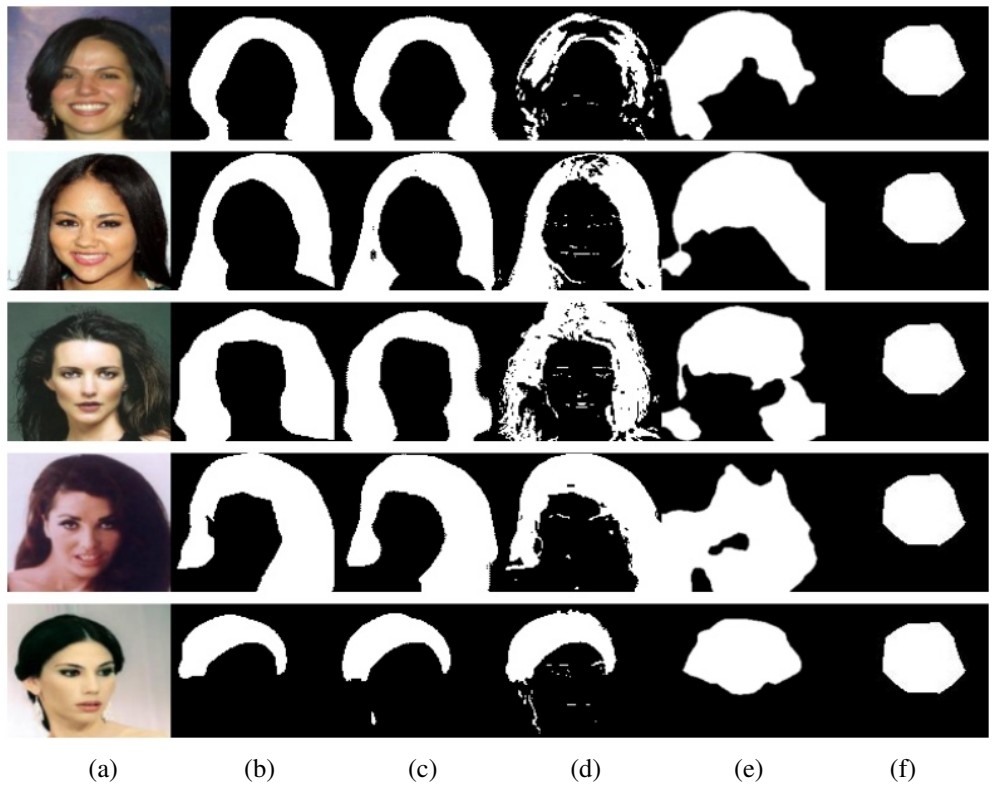

Figure 34: Additional Segmentation results for of women's hair. (a) original image, (b) ground truth segmentation, (c) our results, (d) the results of Press et al. (2019), (e) the results of Ahn & Kwak (2018), (f) results of CAM.

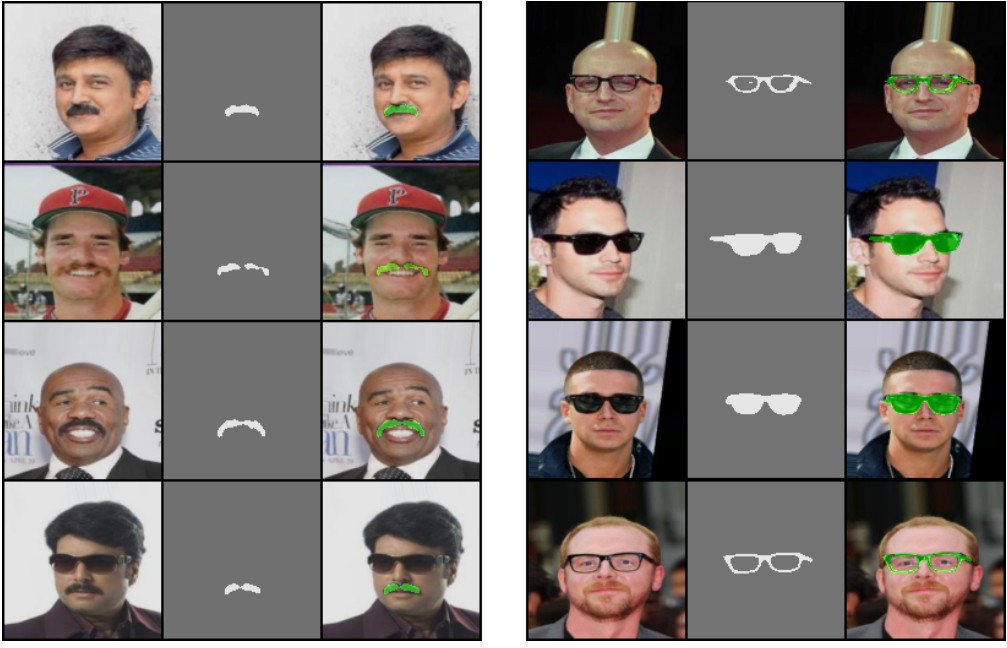

Figure 35: Additional segmentation results for the domain of glasses and facial hair.

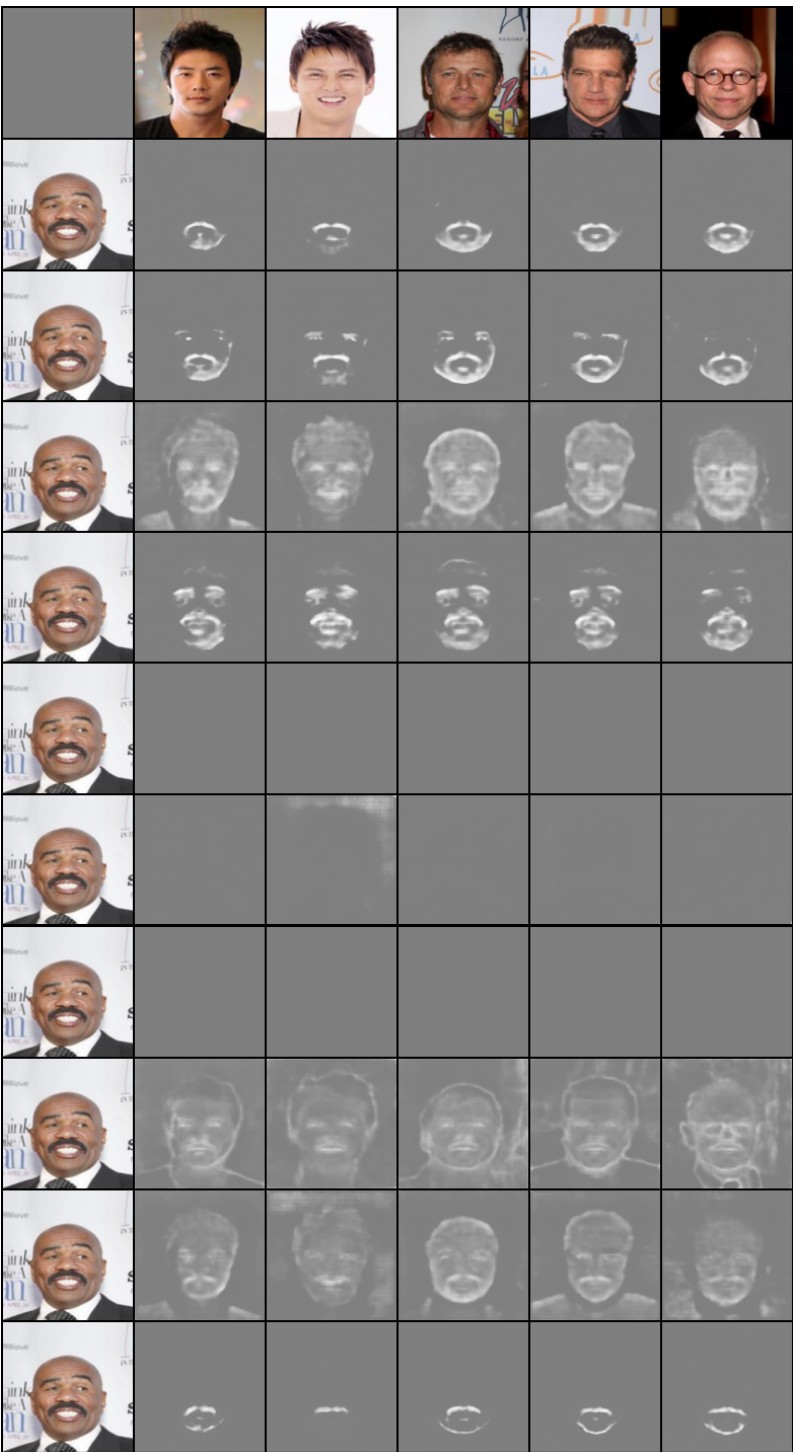

Figure 36: Ablation analysis. The first row is images without facial hair on which we want to transfer the facial hair of the image in the first column. Second row: All losses $\mathcal{L}$, third row: without $\mathcal{L}^A_{Recon2}$, fourth row: without $\mathcal{L}^B_{Recon2}$, fifth row: without $\mathcal{L}_{Cycle}$, sixth row: without $\mathcal{L}^B_{Recon1}$, seventh row: without $\mathcal{L}^A_{Recon1}$, eighth row: without $\mathcal{L}_{DC}$. The ninth row shows the translation where $\mathcal{L}^A_{Recon2}$ and $\mathcal{L}^B_{Recon2}$ are replaced by L2 regularization of the mask. The tenth row: L1 norm is replaced with L2 norm for $\mathcal{L}^A_{Recon2}$ and $\mathcal{L}^B_{Recon2}$. The last row: L1 norm is replaced with L2 norm for $\mathcal{L}^A_{Recon1}$ and $\mathcal{L}^B_{Recon1}$.

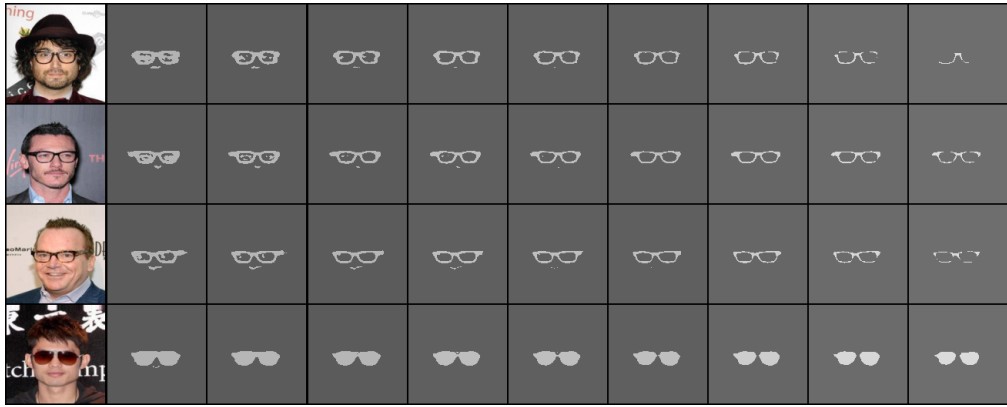

Figure 37: The effect of the threshold used to binarized the mask. The left column is the original image and rest of the column are the segmentation mask created using the thresholds: $0.1, 0.2, 0.3, 0.4, 0.5, 0.6, 0.7, 0.8, 0.9$ (from left to right).

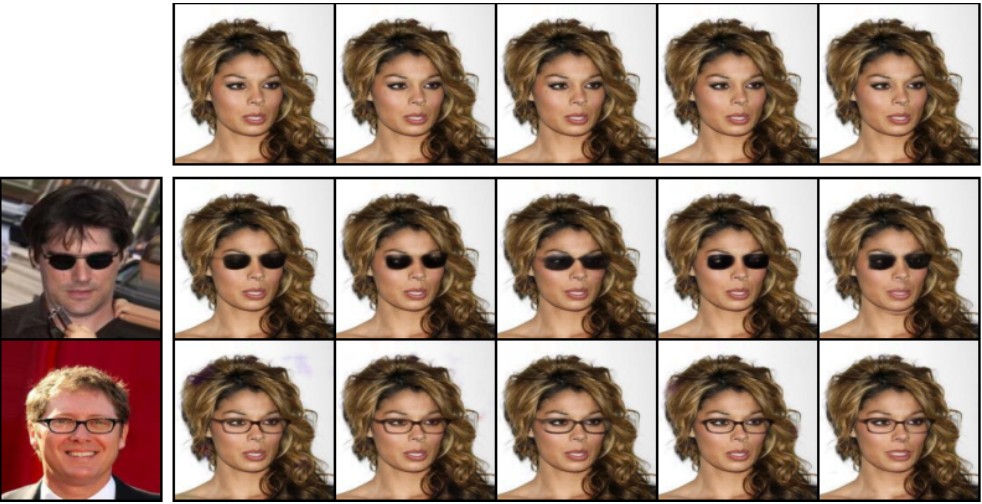

Figure 38: Sensitivity to changes in $\lambda_4$. Given an image with glasses (left), and another image of a face with no glasses (top), the proposed method translates the specified glasses using different values of $\lambda_4$: $0.4, 0.5, 0.7, 0.9, 1.0$ (from left to right). All other hyperparameters remain fixed.

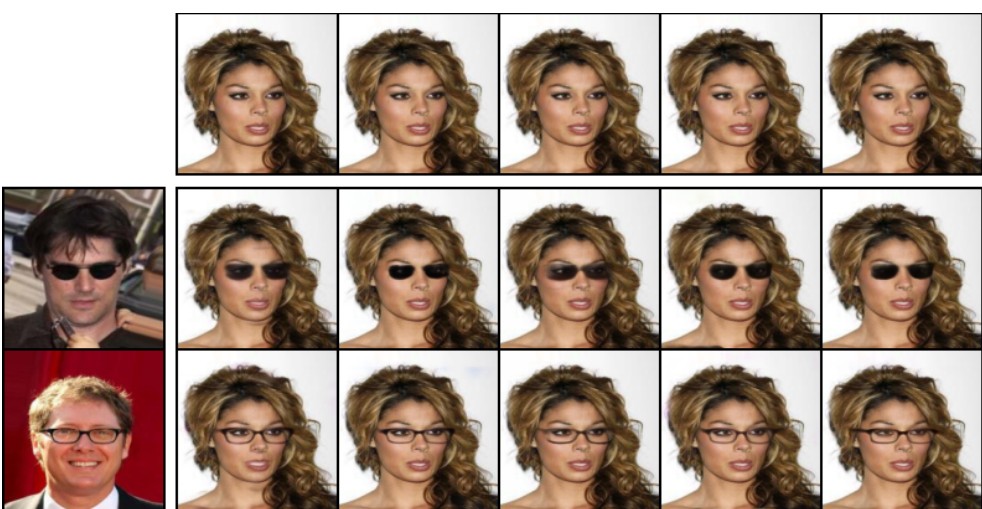

Figure 39: Sensitivity to changes in $\lambda_1$. Given an image with glasses (left), and another image of a face with no glasses (top), the proposed method translates the specified glasses using different values of $\lambda_1$: $3.0, 4.0, 5.0, 6.0, 7.0$ (from left to right). All other hyperparameters remain fixed.

