# OpenReview forum: "Masked Based Unsupervised Content Transfer"
_ICLR.cc/2020/Conference — Accept (Poster)_

### Official Review · AnonReviewer1 · 2019-10-08
**Official Blind Review #1**

**Rating:** 6

**Review:**

This paper proposes a method for unpaired image-to-image translation, where the target domain explicitly contains some additional information than the source domain. The authors use auto-encoders to separate the common and specific representations and to generate masks, which seems to be related to [1]. The authors empirically show the proposed method can be used for image translation, attribute editing.

A small citation error:
"Jun-Yan Zhu, Taesung Park, Phillip Isola, and Alexei A Efros. Unpaired image-to-image translation using cycle-consistent adversarial networkss. arXiv preprint arXiv:1703.10593, 2017a."
networkss -> networks
It is a published paper at ICCV, not just on arxiv.

[1] Domain Separation Networks, Bousmalis et.al, NIPS 2016

**Experience Assessment:**

I do not know much about this area.

**Review Assessment: Checking Correctness Of Derivations And Theory:**

N/A

**Review Assessment: Checking Correctness Of Experiments:**

I assessed the sensibility of the experiments.

**Review Assessment: Thoroughness In Paper Reading:**

I made a quick assessment of this paper.

---

> ### Author Response · Authors · 2019-11-09
> **Response to Review #1**
>
> We thank the reviewer for the comments. We apologize for the citation errors and have corrected these in the revised version.

---

### Official Review · AnonReviewer2 · 2019-10-21
**Official Blind Review #2**

**Rating:** 6

**Review:**

This work proposed a mask based approach for instance-level unsupervised content transfer, which is an extension of the disentanglement work in (Press et al., 2019) and the attention guided translation (Chen et al., 2018, Mejjati et al., 2018). Unlike the disentanglement work, the introduced mask allows the adaptation to focus on the relevant content which substantially reduce the complexity of the generation. On the other hand, the proposed method extends the attention guided translation from the domain level to the instance level which allows more specific and diverse translations. Experiments on benchmark data shows both improved qualitative and quantitative results comparing to existing methods. It is really nice that the authors also considered the situation of generalization to out of domain images.

However, I would encourage the authors to spend more discussion on the "Method" and "Ablation Analysis" sections to give a better illustration. First is the choice of the L2 norm in all the reconstruction losses, which is different from L1 norm used in both (Press et al., 2019) and (Mejjati et al., 2018). What is the advantage of using L2 instead of L1 norm here? Does it work better with the mask generation? Second, the domain confusion loss. The presence of both equation (3) and (4) are quite confusing and the domain confusion loss (3) seems different from traditional ones. In Table 7, it shows that the learned mask is empty without any of the losses (3), (5), (7). But only loss (7) is directly related to the mask generation. How does the loss (3) or (5) impact the mask learning? It is also unclear why the losses introduced in (8) would encourage the mask to be minimal despite the quantitative results shown in Table 7. Actually, I am very curious about the performance of the loss introduced in (Press et al., 2019) on top of the network introduced in Figure 2.

Other comments:
- It would be nice to see the out of domain transfer in the "attribute" domain. Ideally, the network should be able to detect "difference" in the image from domain B and apply it to the image from domain A. For example, the model is trained on faces without and with glasses, but applied to faces without and with facial hair. Indeed, the introduction of mask alleviates the decoder to learn the attribute itself, and provides the ability to locate the place of difference.
- Please unify the citation style: there are both Press et al. (2019) and (Press et al., 2019) used.
- In Section 2 under "Mask Based Approaches", the authors argued that the existing attention guided translation "does not allow for the adaptation of the image information in the masked area". I do not think this is the case. The existing work also introduced adaptation of the image information in the masked area. For example, the equation (1) in (Mejjati et al., 2018).
- How is the binarized mask generated in inference? Specifically, how to determine the threshold?
- In Section 4.1, the authors argued that "without L_{Cycle} the masks produced include larger portions of the face". But this actually produces the second smallest mask in Table 7.
- What is the "L2 reg" in Table 7?
- It would be good to show the sensitivity of the lambdas in the overall loss.

**Experience Assessment:**

I have read many papers in this area.

**Review Assessment: Checking Correctness Of Derivations And Theory:**

N/A

**Review Assessment: Checking Correctness Of Experiments:**

I carefully checked the experiments.

**Review Assessment: Thoroughness In Paper Reading:**

I read the paper thoroughly.

---

> ### Author Response · Authors · 2019-11-09
> **Response to Review #2 Part 1**
>
> We thank the reviewer for the supportive review. We address the comments one by one below. Please let us know if there are any further concerns.
>
> With regards to the choice of L2 instead of L1 for the reconstruction losses, this is an unfortunate typo made in the manuscript, which is now corrected. We used L1 norm for both $L_{Recon2}$ (Eq. 8) and $L_{Recon1}$ (Eq. 5), (Eq. 6) and used L2 norm for the cycle loss (Eq. 9) as can be seen in the supplementary code submitted with the paper.
>
> We added an experiment to the ablation analysis where for loss $L_{Recon1}$ (Eq. 5),(Eq. 6) and loss $L_{Recon2}$ (Eq. 8)  the L1 norm is replaced with the L2 norm. We found the result to be comparable for the losses in Eq. 5 and Eq. 6 and inferior for the one in Eq. 8. Intuitively, since L1 encourages sparsity better than L2, it supports well localized masks. The revised version addresses this in Ablation Analysis section 4.1
>
> Our formulation of the domain confusion loss is similar to that of “Adversarial Discriminative Domain Adaptation” (Tzeng et al. 2017) except where for Eq. 3, $E_c$ (the encoder of the common part) attempts to fool the discriminator, so that the encodings of both domain A and domain B would be classified as 1. Namely, the discriminator tries to distinguish between encodings of domain A and B, while $E_c$ attempts to produce an encoding which is indistinguishable for the discriminator.
>
> While the loss in Eq. 7 directly affects the mask generation, the losses in Eq. 3 and Eq. 5 affect it indirectly. Without the loss in Eq. 3, no disentanglement is possible, and the common encoder would contain all of the image information including the separate information. This means that the image produced by $D_A(E_c(b))$ is close to b and, therefore,  the generated mask is empty.
>
> Furthermore, without the loss of Eq. 5, we empirically observe that $D_A(E_c(b))$ outputs the image with the specific part intact (for example, the facial hair is not removed). This indirect effect on the disentanglement probably stems from the fact that without this loss, there is reconstruction only on faces with facial hair (the specific part). Thus, $E_c$ can encode generic facial hair information for shaved faces and have $E_c(b)$ and $E_c(a)$ still indistinguishable. Eq. 5 makes sure that $E_c$ won’t encode facial hair for shaved faces, since it requires reconstruction of an image without facial hair. This is an interesting phenomenon and is worth investigating as future work.
>
> The loss introduced in the first term of Eq. 8 ($L_{Recon2}^A$) encourages a minimal distance between $z(a,a)$ and $a$, where
>  $z(a,a)= z^{raw}(a,a) \otimes m(a, a) + a\otimes(1-m(a,a))$ . Ideally, $z^{raw}$ would be equal to a, but since we use an encoder and a decoder which cannot auto-encode perfectly, we get that there is some distance between $z^{raw}$ and a. Hence, in order to minimize the distance between $z(a,a)$ and a, the network minimizes the size of the mask. Similar argument holds for $L_{Recon2}^B$.
>
> With regards to running the loss term introduced in (Press et al., 2019) on top of the network introduced in Fig. 2, we found that the network fails in this case. The loss function of (Press et al., 2019) contains a domain confusion term that is equivalent to our domain confusion loss and a reconstruction loss which is $||D(E_c(a)) - a||_1$ + $||D(E_c(b),E_s(b)) - b||_1$. Note that the first term of this reconstruction loss is identical to Eq. 5. The second term needs to be adjusted to a mask formulation, as is done in Eq. 6. As we show quantitatively in the ablation study, without further regularization, the mask produced is very large and the output is of low quality.
>
> The model trained to add glasses would not add a mustache. As future work, it may be interesting to study generalization beyond the attributes seen during training  by learning multiple attributes at once.
>
> Regarding previous methods such as Mejjati et al, we thank the reviewer for this observation. We have changed the text to emphasize that the adaptation of the previous work is not performed on the specific information of a guide image, as is in our case.
>
> With regards to the binary mask threshold, most values are very close to 0 or 1. Early on during the experiments we found the value of 0.6 is visually appealing and we kept it for all experiments. See fig. 38  for the effect of changing this threshold. As can be seen, the change is minimal.
>
> As shown in Tab. 7, there is a 6% difference in the size of the mask when $L_{Cycle}$ is removed. Indeed other losses further affect the size of the mask.
>
> Following the review, the citation style has been corrected in the revised version.
>
> For L2_reg, please refer to the discussion at the end of the ablation analysis of section 4.1 (last 10 lines before last paragraph).

---

> > ### Comment · AnonReviewer2 · 2019-11-14
> > **Thanks for the response**
> >
> > Thanks for the detailed response.
> >
> > The two reconstruction losses introduced in equation (8) basically encourage the model to ignore z^{raw} and generate empty mask to perfectly reconstruct the image using itself. Their effects on the mask size should roughly be the same. But in Table 7, all the metrics are comparable except for the mask size when comparing without L_{recon2}^{A} and without L_{recon2}^{B}. Why does L_{recon2}^{B} have a larger impact on the mask size?

---

> > > ### Author Response · Authors · 2019-11-14
> > > **L_{recon2}^{A} and L_{recon2}^{B}**
> > >
> > > Thank you for your question!
> > >
> > > The mask in z(a,a) is generated using the encodings of the common and specific parts of a, and z(b,b) uses the encodings of the common and specific parts of b. Also, a and b are not symmetrical and since images in A do not contain the specific part, their mask should be minimal.
> > >
> > > The loss L_{Recon2}^A encourages a minimal distance between z(a,a) and a. Similarly, L_{Recon2}^B encourages a minimal distance between z(b,b) and b.
> > >
> > > Therefore, in L_{Recon2}^A the mask is generated from the encoding of an “empty” specific part, while in L_{Recon2}^B, the mask is based on the encoding of a non-trivial specific part.
> > >
> > > In the ablation, the mask difference is evaluated for the case of images from domain B, where the mask should be non-empty (we specify that “an ablation analysis is performed… for the task of facial hair content transfer” and this will be further clarified in the next revision). Therefore, the loss L_{Recon2}^B is both more relevant to the mask being tested and more relevant to the generation non-trivial masks. This makes its impact more substantial.

---

> ### Author Response · Authors · 2019-11-09
> **Response to Review #2 Part 2**
>
> With regards to the sensitivity of the lambda coefficients in our loss, the values of the coefficients were set early on in the development process in a way that reflects the relative importance we attributed to each component and the observed trade-offs. For example, if the mask obtained was too large we would increase $L_{Recon2}$ (Eq. 8). As illustrated in Fig. 39, our network is not overly sensitive to the choice of these values. For example, for $L_{Recon2}^{A}$ (Eq. 8) loss each value in the range 0.4-1.0 results in a similar output and for $L_{Recon1}^{A}$ (Eq. 5) each value in the range 3.0-7.0 results in a similar output.

---

### Official Review · AnonReviewer3 · 2019-11-04
**Official Blind Review #3**

**Rating:** 6

**Review:**

This paper proposes a method to disentangle the common and separate parts of two domains and to focus the attention of the underlying network to the desired part only, without reconstructing the entire target. The proposed method is also able to add or remove separate contents, and to enable weakly-supervised semantic segmentation of the separate part of each domain.
This work relates to the problem of content transfer between images. The proposed method consists of five networks: one encoder for common domain invariant features, one encoder for separate domain specific information, one network for mapping encodings from common features from both domains undistinguishable, a decoder that generates sample in the origin domain and a decoder that generates the image that combines content from the origin image and domain specific content from the target image. This last decoder also outputs a mask that focuses the attention of the model to the specific part.
The proposed model is trained using a combination of different losses: domain confusion loss, reconstruction losses, and cycle consistency losses. Ablation studies reported in the paper nicely show the contribution of each loss. The final loss is obtained by a weighted sum of the losses: how are the lambda coefficient chosen/learned?
The proposed method is evaluated on guided content transfer, out of domain manipulation, attribute removal, sequential content transfer, sequential attribute removal and content addition, weakly supervised segmentation of the domain specific content. Experimental results are clear, thorough and satisfactory, both quantitative and qualitative results are reported, as well as a user study. Presented results demonstrate the strengths and limitations of the proposed approach, and the analysis of the results helps understanding and emphasizing the contribution of the paper.
Information in appendix also enable reproducibility by providing parameters and architecture structure.
Other comments: did you observe overfit for some choice of parameter? is your validation/test set large enough for evaluating results? did you observe biases in your method (e.g. to specific features/domain specific information)?

**Experience Assessment:**

I do not know much about this area.

**Review Assessment: Checking Correctness Of Derivations And Theory:**

I assessed the sensibility of the derivations and theory.

**Review Assessment: Checking Correctness Of Experiments:**

I assessed the sensibility of the experiments.

**Review Assessment: Thoroughness In Paper Reading:**

I read the paper at least twice and used my best judgement in assessing the paper.

---

> ### Author Response · Authors · 2019-11-09
> **Response to Review #3**
>
> We thank the reviewer for the supportive review. We address the comments below. Please let us know if there are any further concerns.
>
> For the choice of lambda coefficients: these were set early on in the development process in a way that reflects their relative importance and were observed. For example, if the mask obtained was too large we would increase $L_{Recon2}$ (Eq. 8).
>
> As illustrated in Fig. 39, our network is not overly sensitive to the choice of these values. For example, for $L_{Recon2}^{A}$ (Eq. 8) each value in the range 0.4-1.0 results in a similar output and for $L_{Recon1}^{A}$ (Eq. 5) each value in the range 3.0-7.0 results in a similar output.
>
> We constructed the train/test sets using 90\%-95\% split (see ablation section A). This consists of about 7,200-18,000 examples for train and about 800-2,000 examples for test for each attribute.
>
> As for the possibility of overfitting, we observed the same performance on the train and test sets, with no noticeable difference between the two.
>
> With regards to potential bias, as far as we observed, there was no bias toward specific shape or appearance, and we did not observe mode-collapse in the form of repeated appearance elements.

---

### Author Response · Authors · 2019-11-09
**A revision following the feedback from the reviewers**

In the revised version, we have corrected the typos pointed to by the reviewers and addressed all requests for elucidations.

In addition, following the reviews we have added the following: First, we added a discussion of the choice and sensitivity of the lambda coefficients including Fig. 39 and Fig. 40 shows the sensitivity of $L_{Recon2}^{A}$ (Eq. 8) and $L_{Recon1}^{A}$ (Eq. 5) losses as well as the choice of the threshold for mask binarization presented in Fig. 38. As the results show, the method is largely robust to its parameters.

Second, we further extend our ablation analysis with regards to the choice of L1 norm vs L2 norm in our losses.

Third, further discussion was added to the Method section and Ablation Analysis section, with regards to the effect of the loss terms in Eq. (3), (5), (7) and (8) and the mask learned. We have also added a discussion with regard to the formulation of the domain confusion loss.

---

### Decision · Program_Chairs · 2019-12-19

**Decision:**

Accept (Poster)

**Comment:**

This paper extends the prior work on disentanglement and attention guided translation to instance-based unsupervised content transfer. The method is somewhat complicated, with five different networks and a multi-component loss function, however the importance of each component appears to be well justified in the ablation study. Overall the reviewers agree that the experimental section is solid and supports the proposed method well. It demonstrates good performance across a number of transfer tasks, including transfer to out-of-domain images, and that the method outperforms the baselines. For these reasons, I recommend the acceptance of this paper.